# A Video Is Not Worth a Thousand Words

## Abstract

As we become increasingly dependent on vision language models (VLMs) to answer questions about the world around us, there is a significant amount of research devoted to increasing both the difficulty of video question answering (VQA) datasets, and the context lengths of the models that they evaluate. The reliance on large language models as backbones has lead to concerns about potential text dominance, and the exploration of interactions between modalities is underdeveloped. How do we measure whether we're heading in the right direction, with the complexity that multi-modal models introduce? We propose a joint method of computing both feature attributions and modality scores based on Shapley values, where both the features and modalities are arbitrarily definable. Using these metrics, we compare 6 VLM models of varying context lengths on 4 representative datasets, focusing on multiple-choice VQA. In particular, we consider video frames and whole textual elements as equal features in the hierarchy, and the multiple-choice VQA task as an interaction between three modalities: video, question and answer. Our results demonstrate a dependence on text and show that the multiple-choice VQA task devolves into a model's ability to ignore distractors.

## 1 Introduction

Since the advent of pre-trained large language models (LLMs) with strong reasoning capabilities, vision language models (VLMs) have rapidly become a catchall system for users desiring to interact with multi-modal models, primarily because they can be queried in a similar manner to humans. VLMs are frequently modelled as a paired vision encoder and a pre-trained LLM, where visual and text tokens are projected into the same input space. With larger and more powerful models, VLMs are now created for visual understanding, either specialising in video or additionally allowing multi-image input. These models place the onus to reason well on the LLM, assuming that so long as the features are well-aligned, the model will understand the relationship between the modalities. We explore this hypothesis by determining the degree that each modality feature contributes to a model's response, and hence, whether modality preferences present themselves in current approaches to video question answering (VQA)—a common benchmark task for video language models.

Our effort stems from an intuition that video is not being reliably integrated into VLMs, motivated by a variety of related work. Recently, Deng et al. (2025) showcased a blind reliance on text when modalities disagree with eachother. Using Shapley values (Shapley, 1997), the modality preference of image/text tasks was investigated by Parcalabescu & Frank (2023; 2025), showing that VLMs are less self-consistent than LLMs. The separate, but related, problem of video redundancy has been investigated by works including Buch et al. (2022), which aimed to determine "what can be understood from a single image", by training a probe to select single frames capable of answering questions intended to be temporally complex. Similarly, Price & Damen (2020) showed that uni-modal video model performance for action recognition could be improved by removing "distractor" frames, calculated via Shapley values.

Where previous works have used token level features for calculating attributions and accuracy-based heuristics for determining modality preference (Parcalabescu & Frank, 2023; Goldshmidt & Horovicz, 2024), we instead propose an attribution method for arbitrarily grouped features that can be used to calculate modality scores. We consider whole frames and text elements (words, numbers, etc.) as individual features, as these are the smallest unit of meaningful information from a human perspective and this incorporates any model specific encoding/tokenisation while reducing the complexity of

the input. Conducting this analysis across a range of VQA datasets, we jointly investigate current capabilities of open source multi-modal models to properly integrate video and text.

In summary, our contributions are as follows: (i) we expand upon feature attribution methods built on Shapley values, being the first to bring it to the video/text domain; (ii) propose a method of calulating modality scores independent of model accuracy and modality length; (iii) benchmark 6 VLM models on 4 datasets; (iv) determine that video as an entire modality is being under-utilised by models—frames are consistently undervalued compared to text and (v) demonstrate that video contribution (and dataset difficulty) can be increased simply by adding more multiple-choice answers.

## 2 RELATED WORK

**Interpretability**    Early interpretability approaches focused on uni-modal models, e.g., vision: (Fan et al., 2022; Koh et al., 2020; Wong et al., 2021; Smilkov et al., 2017), audio (Parekh et al., 2024; Mishra et al., 2017; Becker et al., 2018), and language: (Ribeiro et al., 2016a; Bau et al., 2017; Chen et al., 2024a; Shi et al., 2023; Xie et al., 2024b). Works focused on gradient based methods (Sundararajan et al., 2017; Binder et al., 2016; Selvaraju et al., 2017) and attention-based methods concurrent with the rise of transformers (Jain & Wallace, 2019; Wiegreffe & Pinter, 2019). Others employed feature attribution methods (Goldshmidt & Horovicz, 2024; Price & Damen, 2020) using Shapley values (Shapley, 1997) from game theory, which calculates the contribution of players within a co-op game. Shapley values have been employed as a black box approach, requiring only perturbations to the input and examining the output to interpret the model. Specifically, SHapley Additive exPlanation (SHAP) (Lundberg & Lee, 2017), unified Shapley values such that given a prediction, each input feature is assigned an importance value.

With the rise in popularity of vision language models, recent work has also focused on interpretability across multi-modal models (Stan et al., 2024; Chen et al., 2024d; Aflalo et al., 2022; Chefer et al., 2021; Liu et al., 2024; Ramesh & Koh, 2022; Swamy et al., 2023). DIME (Lyu et al., 2022)—itself an extension of LIME (Ribeiro et al., 2016b)—interprets models via the pertubation of input features to affect the model outputs. It first disentangles the model into uni-modal and multi-modal contributions before generating intepretable visualisations via applying LIME. Similar to our work, MM-SHAP (Parcalabescu & Frank, 2023) extends SHAP for multiple modalities, in this case for image and text, to measure contributions of image patches and words in text to the final decision made by the model. PixelSHAP (Goldshmidt, 2025), was proposed to instead model the contributions of feature groups of pixels related to the same concept within the input image, thus allowing for a finer granularity than attributions of image patches. Our work lies within the feature attribution paradigm, extending SHAP for the task of VQA enabling both frame attribution and modality dependence for video, question, and answer inputs.

**Modality Preference**    Concurrent with interpretability approaches, Huang et al. (2022) theorise that "During joint training, multiple modalities will compete with each other [and only a subset of modalities will be utilised] ... with other modalities failing to be explored." Many works have reached similar conclusions by exploring modality dominance. These works have focused on: pertubation of input modalities (Park et al., 2025; Wu et al., 2022)/input features (Parcalabescu & Frank, 2023; Frank et al., 2021) to show that one modality is dominant; finding similar images which showcase the gap between vision and vision-language (Tong et al., 2024); few shot evaluation of language models (Chen et al., 2024b); or hard negatives/foiling examples (Parcalabescu et al., 2022; Gat et al., 2021; Deng et al., 2025). Previous work has overwhelmingly shown that models frequently exhibit a preference to one or more modalities, which is commonly language within a vision language model. Others have proposed solutions to the modality bias, typically by introducing datasets that require models to understand and utilise information from both modalities (Goyal et al., 2017; Parcalabescu et al., 2022; Chen et al., 2024c) or via training regimes that balance the utilisation of each modality (Yang et al., 2024c; Leng et al., 2024; Wang et al., 2024a; Xiao et al., 2025; Pi et al., 2024; Deng et al., 2024). Closest to our work, Park et al. (2025) propose the Modality Importance Score (MIS) to discover modality bias for VQA, showing that current models do not effectively utilise information across multiple modalities. They conclude that the datasets they evaluated for VQA do not demand multi-modal reasoning, with 90–95% of questions requiring only a single modality or are modality agnostic. Our metrics differ from MIS in two ways: Firstly, our metrics can provide feature level attribution *in addition* to modality preference. Secondly, our metrics can provide the degree at which a feature is positive/negative/neutral rather than as ternary outcomes.

# 3 METHOD

## 3.1 SHAPLEY VALUES

Typically, deep learning models take a large set of input features as input and distill them into a one-dimensional confidence (or logit) in any given output class (or token), making it exceptionally difficult to identify the degree of impact that each feature has. Shapley values were introduced to the field of game theory in Shapley (1997), as a solution concept for the fair distribution of contributions to a cooperative game. More formally, given a set of $N = \{1, 2, \ldots, n\}$ players, the function $r : 2^N \to \mathbb{R}$ maps from all possible subsets of $N$ (the power set) to a value (or reward). If a coalition of players is represented as $S$, then $r(S)$ represents the total contribution of that coalition.

**Definition 3.1 (The Shapley value)** *The Shapley value of player $i$, $\varphi_i(r)$ can be defined as:*

$$\varphi_i(r) = \frac{1}{n!} \sum_{P \in S_n} \left( r(P_i \cup i) - r(P_i) \right)$$

*where $S_n$ is the set of all possible permutations (or orderings) of $N$ and $P_i$ is the set of players that precede player $i$ in a given permutation.*

Informally, this formula represents taking every possible order that the $N$ features can be added to the coalition and averaging the marginal contributions of the $i$th feature.

Shapley values are the only payment rule to satisfy the following four properties of: **Efficiency**: the sum of contributions across all players is equal to the model output; **Symmetry**: if two players contribute equally then they will be assigned an equal score; **Linearity**: the Shapley value of the sum of two (or more) reward functions is the sum of their individual Shapley values; and **Null Player**: if a player does not contribute then it will be assigned a score of 0.

It should be clear that Shapley values are a convincing candidate as a method to compute local feature attributions of a model, by which we mean: the impact that the features of a single input instance will have on a model's prediction. We now consult the work of Lundberg & Lee (2017) to provide a unified framework for using Shapley values as an attribution method for any arbitrary machine learning model. Assuming we have some prediction model $f$, we define a new explanation model $g$, which takes a simplified input $x'$. This simplified input is mapped to the real input $x$ by some mapping function $h_x(\cdot)$, depending on $x$.

**Definition 3.2 (Additive feature attribution)** *An additive feature attribution method has explanation model:*

$$g(x') = \phi_0 + \sum_{i=1}^{M} (\phi_i \times x'_i)$$

*where $x' \in \{0, 1\}^M$, $M$ is the number of simplified features and $\phi_i \in \mathbb{R}$ is the attribution of each simplified feature.*

Through this definition, we can arbitrarily group a model's features at any scale, e.g., images instead of pixels or words instead of text tokens. Zeros in the simplified feature vector $x'$ represent masking the entire grouped feature out, and ones represent keeping the full grouped feature in the input. Subsequently, the following three properties are desirable for an additive feature attribution method: **Local Accuracy**: when the original model is approximated, its output should match the original output; **Missingness**: if the feature is missing it won't be given any attribution; and **Consistency**: if a feature's contribution is constant/increases due to a model change, it's attibution won't decrease. According to Lundberg & Lee (2017) the attribution satisfying these properties, is the Shapley value.

## 3.2 MULTI-MODAL SHAPLEY VALUES

In our case, the "game" is multiple-choice VQA, and each "player" is a frame or word in the input. Calculating exact Shapley values is computationally infeasible for large numbers of features, as the number of possible coalitions scales in a factorial manner, therefore it is typically approximated. In this paper, we use a Monte Carlo method to approximate Shapley values from the SHAP library (Lundberg, 2018).

We'll refer to the VLM model as $f$ (whose logits will be used as reward $r$), taking multi-modal video and text input $(v, t)$, which generates an arbitrary number of text tokens. The video modality $v$ is a sequence of $n_v$ video frames (or video features) and the text modality $t$ is a sequence of $n_t$ textual elements. Textual elements include individual words, numbers or punctuation. We choose to represent atomic elements rather than sub-word tokens to match human perspective. We can transform the multi-modal features $(v, t)$ to simplified features $x' \in \{0, 1\}^{n_v + n_t}$ by defining a mapping function that retains the multi-modal feature $(v, t)_i$ if $x'_i = 1$ and masks it out if $x'_i = 0$. In the video modality, by masking we simply zero out all of the pixels/features of the masked image. Conversely, in the text modality, masking is represented by changing the textual element to be whitespace.

To update the Shapley values, we need to retrieve a valid reward from the model in relation to the input. We opt to use the logits of the predicted text tokens, as it does not require internal information from the model and can be returned along with these tokens. In multiple-choice VQA, each question has $n_c$ corresponding choices (or classes) for the model to pick from, where we can jointly calculate the Shapley values for all classes by simply retrieving the respective logits. We label the answers from A–E, skipping letters if the question does not provide 5 choices and query the model to pick a letter. In cases where the language model does not respect requests to respond with only a single letter token, we first check whether a letter from A–E appears in the output. If it does, we index this token, otherwise we take the first token in the output sequence.

## 3.3 METRICS

In this subsection we'll explain the quantitative metrics that we'll use to highlight differences between the Shapley values for the question modalities: video (V), question (Q), and answers (A)[1]. The reason for differentiating between questions and answers arises from an intuition that in multiple-choice, the semantic purpose of the questions is different enough to that of the answers. Given a model and a dataset of VQA-tuples—containing a video, a question, and the corresponding answers—these metrics are calculated over the Shapley values for all of these tuples. While the Shapley values are divided by class $c$, in practice we'd like to split them based on whether they are the ground truth or negative. Let $\varphi_i^{\text{gt}} \in \mathbb{R}$ refer to the Shapley value of multi-modal element $i$ for the ground truth class and $\varphi^{\text{gt}}$ be the list of these Shapley values. Furthermore, let $n_v$, $n_q$ and $n_a$ be the number of video, question and answer elements respectively. The scale of a model's logits can vary, so to normalise a VQA-tuple's Shapley values, we divide by the maximum absolute value to normalise the values in the interval of $[-1, 1]$ while, importantly, preserving 0 values: $\hat{\varphi}_i^c = \varphi_i^c / |\max_i \varphi_i^c|$.

The following metrics are defined per VQA-tuple and will be averaged over the dataset. For our first quantitative metric, we want to measure the magnitude of *contribution* of each VQA-tuple modality, which we call Modality Contribution (MC). We calculate this contribution by dividing the total magnitude per modality by the sum of the total magnitudes of each modality. Consequently, the modality specific contributions are the proportion of this total:

$$\text{MC}_\text{V} = \frac{\sum_{i=1}^{n_v} |\hat{\varphi}_i^{\text{gt}}|}{\sum |\hat{\varphi}_i^{\text{gt}}|}, \quad \text{MC}_\text{Q} = \frac{\sum_{i=n_v+1}^{n_v+n_q} |\hat{\varphi}_i^{\text{gt}}|}{\sum |\hat{\varphi}_i^{\text{gt}}|}, \quad \text{MC}_\text{A} = \frac{\sum_{i=n_v+n_q+1}^{n_v+n_q+n_a} |\hat{\varphi}_i^{\text{gt}}|}{\sum |\hat{\varphi}_i^{\text{gt}}|}$$

The second quantitative metric measures the *average contribution* of each feature within each modality, which we refer to as Per-Feature Contribution (PFC). When the number of features in a modality is significantly high, that modality can become over-represented, leading to a large sum of many smaller Shapley values. Taking the mean of the magnitudes per modality helps to avoid the imbalance caused by differing numbers of features. To simplify the formula, we define the halfway variable $M$ to represent the mean Shapley value of a modality:

$$M_\text{V} = \frac{\sum_{i=1}^{n_v} |\hat{\varphi}_i^{\text{gt}}|}{n_v}, \quad M_\text{Q} = \frac{\sum_{i=n_v+1}^{n_v+n_q} |\hat{\varphi}_i^{\text{gt}}|}{n_q}, \quad M_\text{A} = \frac{\sum_{i=n_v+n_q+1}^{n_v+n_q+n_a} |\hat{\varphi}_i^{\text{gt}}|}{n_a}$$

Then the Per-Feature Contributions are the proportion of these summed means:

$$\text{PFC}_\text{V} = \frac{M_\text{V}}{M_\text{V} + M_\text{Q} + M_\text{A}}, \quad \text{PFC}_\text{Q} = \frac{M_\text{Q}}{M_\text{V} + M_\text{Q} + M_\text{A}}, \quad \text{PFC}_\text{A} = \frac{M_\text{A}}{M_\text{V} + M_\text{Q} + M_\text{A}}$$

---

[1]We assume for this paper that we have 3 modalities in a specific order, but these metrics extend trivially to any number of modalities.

## 4 RESULTS

### 4.1 MODELS AND DATASETS

**Models**     We choose 6 VLM models to examine using our metrics defined previously to cover several aspects: Firstly, to have a variety of context lengths. Secondly, to compare two-stream encoder approaches to VLMs which use LLM decoders. Thirdly, to investigate how models might have changed over time, evaluating older vs. newer approaches. According to these requirements, we select the following: **FrozenBiLM** (Yang et al., 2022): A compact, early example of a two-stream VLM model that uses pre-extracted CLIP (Radford et al., 2021) features. We use it with 10 frames. **InternVideo** (Wang et al., 2022): An early two-stream foundation video model. We use it, again, with 10 frames. **VideoLLaMA2** (Cheng et al., 2024): A short context VLM using Mistral (Jiang et al., 2023) as the LLM decoder designed for question answering/captioning and to improve spatio-temporal reasoning. We use the maximum of 16 frames. **LLaVA-Video** (Zhang et al., 2024): A medium context VLM using Qwen2 (Yang et al., 2024a) as the LLM decoder, trained with a curated dataset and carefully intruction tuned. We use 64 frames. **LongVA** (Zhang et al., 2025b): A long context open-source model, also using Qwen2 as the LLM decoder. We use 128 frames (the maximum that we could fit into GPU memory). **VideoLLaMA3** (Zhang et al., 2025a): An update to VideoLLaMA2 using Qwen2.5-7B (Yang et al., 2024b) as the LLM decoder, trained on a longer context with a focus on efficiency. We used the maximum of 180 frames.

**Datasets**     We select 4 datasets to evaluate under the following requirements; Firstly, to cover both popular and new datasets. Secondly, showcasing first-person (egocentric) and third-person (exocentric) viewpoints and, finally, to include long and short video contexts. We took subsets (necessary due to the excessive time it would take to calculate Shapley values for entire datasets) from the following 4 VQA datasets: **EgoSchema** (Mangalam et al., 2023): Egocentric VQA dataset intended to be unanswerable without viewing all of the video. We take a subset of 50 questions from the set released with ground truths. **HD-EPIC** (Perrett et al., 2025): Egocentric VQA dataset collected within the kitchen domain, ranging from questions about single images to multi-video queries spanning several hours. We take a subset of 60 questions, with 2 from each of the 30 question types. **MVBench** (Li et al., 2024): General VQA dataset curated from several other datasets to create a multiple-choice benchmark across different tasks. We take a subset of 60 questions, 3 from each question type. **LVBench** (Wang et al., 2024b): General VQA dataset for very long video understanding, where many questions are asked about small collection of lengthy YouTube videos. We take a subset of 60 questions, 10 from each question type.

### 4.2 WHAT DOES THE CONTRIBUTION METRIC SHOW?

In table 1, we first show the Modality Contribution and Per-Feature Contribution for each model/-dataset combination along with the corresponding accuracy. The cells are highlighted to show high (**blue**) and low (**red**) contribution scores. Note, for all of these coloured tables, values of $1/3$ would represent balanced contributions across the three modalities.

**Under-representation of Video**     For all methods but VideoLLaMA3, video is consistently under-represented in the Modality Contribution, indicating that the modality as a whole is consistently contributing less to the decision of the models. VideoLLaMA3 on the other hand, shows strong contributions from video, especially for LVBench, where the entire dataset comprises of $\sim 1$ hour long videos. Looking at the Per-Feature Contribution values, we see that video is still consistently underrepresented among the three modalities. However, for long context models, video shows vastly reduced contributions, meaning that per-frame the Shapley values are much smaller than their text feature counterparts. Video as a whole modality is clearly still highly relevant, but this is evidence that the Shapley values of its individual frames are more centered around zero, and that the model's attention to them is much less guided than for the text.

**Importance of Question vs. Answer**     The question is particularly important for FrozenBiLM and InternVideo because the answers are independently queried as multiple binary questions for these models—they do not need to discriminate between answers. For the stronger models, the question is often undervalued compared to the answers, indicating that the model cares less about the specifics of the question, and more about discriminating between the possible answers. This follows recent design of VQA datasets to use hard negatives answers to ensure results are not text-biased, i.e., Perrett et al. (2025); Xie et al. (2024a); Chen et al. (2024c).

Table 1: MC and PFC for each modality in the VQA-tuple. Calculated based on the Shapley values for the ground truth logit averaged across all VQA-tuples. Here **blue** scores relate to large magnitudes of Shapley values, regardless of their sign, while **red** scores relate to values close to 0.

(a) EgoSchema

| | Modality Contribution | | | Per-Feature Contribution | | | Acc |
|------|------|------|------|------|------|------|------|
| | V | Q | A | V | Q | A | |
| FBLM | 0.09 | 0.33 | 0.58 | 0.31 | 0.46 | 0.23 | 0.20 |
| IV | 0.20 | 0.36 | 0.44 | 0.51 | 0.36 | 0.13 | 0.36 |
| VL2 | 0.11 | 0.18 | 0.71 | 0.30 | 0.33 | 0.36 | 0.56 |
| L-V | 0.15 | 0.15 | 0.70 | 0.16 | 0.36 | 0.48 | 0.72 |
| LVA | 0.12 | 0.13 | 0.75 | 0.07 | 0.36 | 0.57 | 0.48 |
| VL3 | 0.30 | 0.14 | 0.56 | 0.14 | 0.40 | 0.46 | 0.70 |

(b) HD-EPIC

| | Modality Contribution | | | Per-Feature Contribution | | | Acc |
|------|------|------|------|------|------|------|------|
| | V | Q | A | V | Q | A | |
| FBLM | 0.09 | 0.56 | 0.36 | 0.19 | 0.56 | 0.24 | 0.22 |
| IV | 0.34 | 0.51 | 0.15 | 0.55 | 0.39 | 0.07 | 0.15 |
| VL2 | 0.17 | 0.25 | 0.58 | 0.27 | 0.26 | 0.47 | 0.28 |
| L-V | 0.19 | 0.26 | 0.55 | 0.17 | 0.32 | 0.51 | 0.35 |
| LVA | 0.18 | 0.22 | 0.61 | 0.13 | 0.28 | 0.59 | 0.35 |
| VL3 | 0.36 | 0.21 | 0.43 | 0.17 | 0.34 | 0.48 | 0.35 |

(c) MVBench

| | Modality Contribution | | | Per-Feature Contribution | | | Acc |
|------|------|------|------|------|------|------|------|
| | V | Q | A | V | Q | A | |
| FBLM | 0.13 | 0.49 | 0.38 | 0.16 | 0.47 | 0.37 | 0.40 |
| IV | 0.26 | 0.51 | 0.23 | 0.32 | 0.48 | 0.20 | 0.42 |
| VL2 | 0.19 | 0.27 | 0.54 | 0.17 | 0.26 | 0.57 | 0.62 |
| L-V | 0.23 | 0.26 | 0.52 | 0.06 | 0.30 | 0.64 | 0.65 |
| LVA | 0.14 | 0.24 | 0.63 | 0.02 | 0.27 | 0.71 | 0.45 |
| VL3 | 0.40 | 0.26 | 0.34 | 0.05 | 0.41 | 0.54 | 0.65 |

(d) LVBench

| | Modality Contribution | | | Per-Feature Contribution | | | Acc |
|------|------|------|------|------|------|------|------|
| | V | Q | A | V | Q | A | |
| FBLM | 0.17 | 0.44 | 0.40 | 0.23 | 0.46 | 0.32 | 0.30 |
| IV | 0.34 | 0.44 | 0.22 | 0.42 | 0.43 | 0.15 | 0.25 |
| VL2 | 0.23 | 0.23 | 0.54 | 0.23 | 0.27 | 0.50 | 0.27 |
| L-V | 0.26 | 0.21 | 0.53 | 0.09 | 0.30 | 0.61 | 0.42 |
| LVA | 0.30 | 0.17 | 0.52 | 0.05 | 0.29 | 0.66 | 0.35 |
| VL3 | 0.47 | 0.16 | 0.37 | 0.08 | 0.34 | 0.58 | 0.48 |

**Dataset Comparisons** According to the Per-Feature Contribution, across the datasets, video is consistently more important for EgoSchema and HD-EPIC than it is for LVBench and MVBench. The video content of these two egocentric datasets is much more diverse (i.e. camera pose, occlusion and complexity of actions/sequences) than for exocentric datasets, evidenced by the relative increase in each frame's importance. Answer Per-Feature Contributions are particularly large for MVBench and LVBench because the answers are often shorter for these datasets, leading to a high density of relevant information, whereas EgoSchema's whole sentences are more akin to natural language and contain more realistic textual distractors, present in its high answer Modality Contributions.

## 4.3 HOW DOES MASKING INPUT AFFECT ACCURACY?

In table 2, we compare the accuracy of vanilla input, no input and masked modalities. The datasets are generally not completely answerable blind, i.e. with the video modality being masked, which reassures the intuition that these benchmarks have been developed to not be trivial for a language-only model to solve. However, it's possible to get between $30\%$ and $50\%$ on EgoSchema and MVBench without video—well above the random performance of $20\%$ and $29\%$—meaning that the combination of the question and answers is pushing the model towards the ground truth. Masking out the question has a surprisingly small effect on performance, but this follows from the low question Modality Contribution values presented earlier, as it is frequently undervalued. HD-EPIC is the hardest to answer without question, suggesting its complexity is important, as the question content of the other datasets is usually less specialised to a specific domain and much shorter. Overall, the under-reliance on question leads us to believe this is clear evidence of a basic limitation of multiple-choice VQA: with just the video and the answer the model appears to be able to discriminate between the answers. In particular, sometimes a model will gain performance when the question is completely removed, suggesting that the task is less about truthfully meeting the requirements of a question, and more about picking amongst a set of information-dense answers in a similar fashion to a video-text matching task. As one would expect, masking out the answer generally gives the largest drop in performance compared to the other modalities with many models being reduced to random performance.

## 4.4 ANSWER REPLACEMENT

To determine the extent to which these trends are biased by the number of answers typically used in multiple-choice questions, we experiment with injected negatives in fig. 1, by automatically creating

Table 2: *How does masking modalities affect performance?* We mask either all input features ("All"), or each modality separately and compare to baseline performance. "None" represents the vanilla accuracy. Here **green**/**red** refers to an increase/decrease in accuracy compared to baseline.

| Model | Masking | EgoSchema | HD-EPIC | MVBench | LVBench |
|---|---|---|---|---|---|
| FrozenBiLM | None | 0.20 | 0.22 | 0.40 | 0.30 |
| | All | -0.02 | +0.00 | -0.23 | +0.03 |
| | Video | +0.00 | -0.03 | -0.05 | -0.02 |
| | Question | +0.02 | -0.02 | -0.03 | +0.00 |
| | Answer | -0.02 | +0.00 | -0.23 | +0.03 |
| InternVideo | None | 0.36 | 0.15 | 0.42 | 0.25 |
| | All | -0.18 | +0.07 | -0.25 | +0.08 |
| | Video | -0.14 | +0.07 | -0.10 | +0.05 |
| | Question | +0.06 | +0.02 | +0.02 | +0.03 |
| | Answer | -0.18 | +0.07 | -0.25 | +0.08 |
| VideoLLaMA2 | None | 0.56 | 0.28 | 0.62 | 0.27 |
| | All | -0.36 | +0.00 | -0.30 | -0.02 |
| | Video | -0.18 | +0.00 | -0.23 | +0.00 |
| | Question | +0.02 | -0.13 | -0.13 | +0.05 |
| | Answer | -0.34 | +0.03 | -0.22 | -0.03 |
| LLaVA-Video | None | 0.72 | 0.35 | 0.65 | 0.42 |
| | All | -0.54 | -0.17 | -0.48 | -0.08 |
| | Video | -0.26 | -0.12 | -0.23 | -0.07 |
| | Question | +0.04 | -0.05 | -0.07 | -0.12 |
| | Answer | -0.58 | -0.17 | -0.28 | -0.32 |
| LongVA | None | 0.48 | 0.35 | 0.45 | 0.35 |
| | All | -0.28 | -0.17 | -0.15 | -0.05 |
| | Video | -0.10 | -0.08 | -0.05 | -0.05 |
| | Question | -0.06 | -0.12 | -0.02 | +0.05 |
| | Answer | -0.30 | -0.17 | -0.12 | -0.17 |
| VideoLLaMA3 | None | 0.70 | 0.35 | 0.65 | 0.48 |
| | All | -0.52 | -0.23 | -0.48 | -0.15 |
| | Video | -0.26 | -0.05 | -0.33 | -0.20 |
| | Question | -0.06 | -0.07 | -0.08 | -0.08 |
| | Answer | -0.48 | -0.23 | -0.40 | -0.18 |

new annotations for VQA-tuples. We introduce two types of answer replacement: "Easy" where the negatives are simply rotated in from one randomly selected question, and "New-$x$" where $x$ new negatives are randomly added from across questions (increasing the number of options) and positions are shuffled. MVBench is excluded as it does not fix the number of options, and "New" negatives are only valid for HD-EPIC and LVBench if the question type is the same as the original negatives.

When testing the "Easy" negatives, the accuracy rises drastically as it becomes easy for the model to match context between the positive answer and the question. As the number of extra answers increases from 5 to 20 in the "New" case, the contribution of answer features decreases while the contribution of video and question features increases. To see whether this translates into model performance, we include modality masking tables in appendix E.8. For example, with VideoLLaMA3 (when adding 10 answers and masking video) that performance drops by 40% and 15% on EgoSchema and LVBench respectively, while when masking questions the drop is 6% and 18% respectively. This indicates that merely adding more multiple-choice answers can drastically alter the contribution of under-represented modalities. Finally, when evaluating model performance on these new VQA-tuples, we see that the accuracy consistently decreases until "New-15" where the drop is often less pronounced.

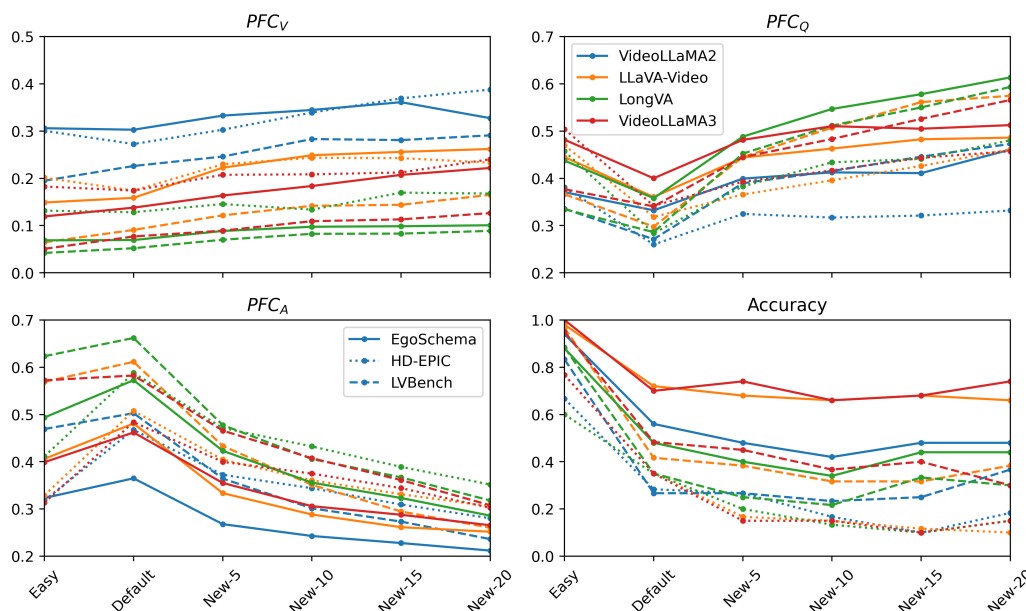

Figure 1: Per-Feature Contribution and accuracy as new negative answers are injected into the VQA-tuples, varying from easiest to hardest.

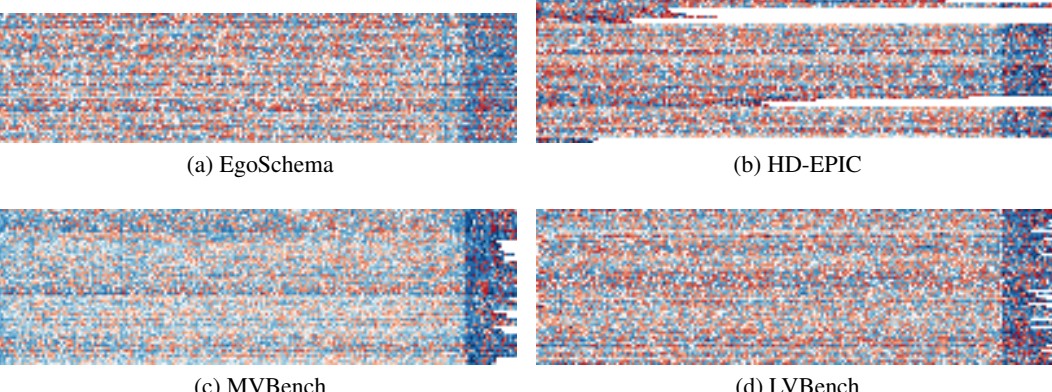

Figure 2: Matrix of Shapley values per subset, where each row, left-to-right, represents the features of a VQA-tuple. Rows are truncated to a maximum of 200 features.

### 4.5 QUALITATIVE RESULTS

We visualise the overall distribution of attributions for each VQA-tuple (row) using VideoLLaMA3, in fig. 2. We truncate this figure for readability given the variable length of the questions and answers. The magnitude of the Shapley values are much larger towards the right hand side of each heatmap, representing the question and answer attributions. This stark boundary is where the video frames end and the text features begin, demonstrating that the video modality contribution is much less than the question/answer. Whilst there are many peaks in the questions and answers, most frames within the video tend to have a similar contribution. Reassuringly, the values for the video frame attributions do not show strong temporal bias, i.e., there are no similar values within each column, yet they are not unstructured, with many examples of several temporally consecutive frames having similar signs. However, MVBench video is typically more positive and slightly skewed to the first frame, showing that latter parts of the video do not contribute as much as in the other datasets.

Next, we provide an example VQA-tuple from EgoSchema evaluated using VideoLLaMA3 in fig. 3 (more examples in appendix F.2). Even though the displayed frames are the 16 most relevant, we

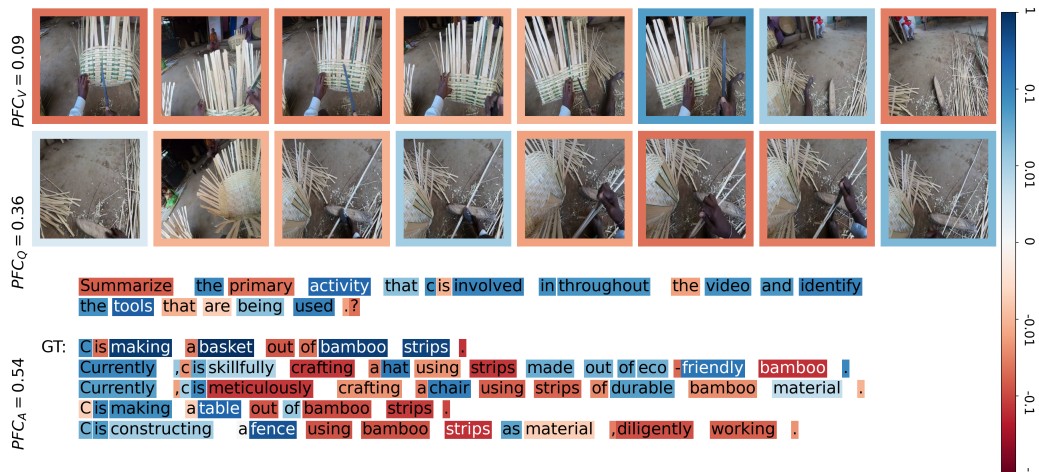

Figure 3: Qualitative figure of an example from EgoSchema evaluated using VideoLLaMA3. For brevity, we select the 16 most important frames, ranked by the magnitude of their Shapley values. Here **blue** represents positively attributed inputs whereas **red** represents negatively attributed inputs.

note that the frames show much smaller contributions, either **positively** or **negatively**, compared to the questions and the answers. This can be seen in the contribution scores, with $PFC_V = 0.09$, whereas $PFC_Q = 0.36$ and $PFC_A = 0.54$. We find no strong trend between +ve/-ve attribution score of each frame and its contents—frames depicting extremely similar content (e.g., the 5th and 6th or 12th and 14th frames) can vary greatly. Interestingly, the relevant nouns that vary in the false answers ("hat", "chair", "table" and "fence") all pull the model towards the ground truth, while "bamboo strips" (appearing in all answers) pushes away from the ground truth, providing evidence of discriminators in the text being employed to choose amongst multiple-choice answers.

## 5   LIMITATIONS AND FUTURE WORK

Whilst within this work we aim for a thorough suite of benchmark datasets (of which we could only use subsets) and VLMs, we recognise this is not an exhaustive collection of both, which could be expanded for future work. As the number of coalitions to calculate true Shapley values scales factorially and we have limited computational resources, we approximate them for our results, using 5000 coalitions across all experiments (ablated in appendix C). As well as this, we focus entirely on multiple-choice VQA so that Shapley contributions are based on reward from the ground truth instead of the generated token, where it would be interesting if it is possible to study open-ended VQA in the same manner. Pushing this interpretability framework further by including more modalities like audio or 3D data will be a relevant focus for future research as multi-modal models grow.

## 6   CONCLUSION

In this paper, we have proposed a joint method of both feature attribution and modality scoring based on the Shapley values of VLMs for VQA. In general, our metrics indicate that VLMs under-represent video compared to the question or answers. By defining a modality metric normalised by the modality length, we find that there is significant divergence between individual frame and text contributions. Furthermore, we demonstrated that the VQA task is limited in properly evaluating multi-modal understanding, as strong accuracy can be achieved without even being presented with a question. We also use the framework to explore a scenario where trivially adding multiple-choice options *beyond* the typical $4/5$ improves dataset difficulty, and find corresponding increases in video feature contribution and dependence of the video modality as a consequence. We hope that our paradigm can be used to better understand and develop future multi-modal understanding in a flexible and interpretable manner, whether it be to examine inputs case-by-case, benchmark entire datasets or test whether models are reliable, unbiased by their impressive improvements in accuracy.

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

## APPENDIX

In the technical appendices we provide further related work for Video Question Answering in appendix A; more details about the mathematical background of Shapley values in appendix B; additional implementation details in appendix C; statistics of each of the dataset subsets in appendix D; further quantitative results in appendix E; and qualitative results in appendix F.

## A    FURTHER RELATED WORK

**Video Question Answering**     Video question answering (VQA) was an extension of the visual question answering task (Antol et al., 2015), from images to videos (Tapaswi et al., 2016; Zhu et al., 2017). Initially, VQA methods utilised a two-stream encoder approach (Zha et al., 2019; Ye et al., 2017; Yang et al., 2020; Seo et al., 2021; Park et al., 2021; Li et al., 2019; Kim et al., 2021; 2020; Jiang & Han, 2020; Jiang et al., 2020; Huang et al., 2020; Gao et al., 2018; Fan et al., 2019; Dang et al., 2021; Wang et al., 2024c) to answer questions, which has progressed to multi-modal large language models (MLLMs) able to reason further about the visual and textual content (Ye et al., 2025; Li et al., 2023; Maaz et al., 2024; Ye et al., 2023). Recently, VQA has become a common benchmark for MLLMs with many datasets being recently proposed to test various aspects of methods' understanding. There has been a growing trend among VQA datasets and methods on longer videos (Wang et al., 2024b; Wu et al., 2024; Song et al., 2024; Fang et al., 2024) as well as constructing datasets that require strong multi-modal understanding (Xiao et al., 2021; Perrett et al., 2025; Mangalam et al., 2023; Chen et al., 2024c;b; Xie et al., 2024a). In this work, we investigate 6 VLMs, ranging from two-stream approaches (Yang et al., 2022; Wang et al., 2022) to MLLMs (Cheng et al., 2024; Zhang et al., 2024; 2025b;a) across 4 recent, challenging datasets (Perrett et al., 2025; Mangalam et al., 2023; Wang et al., 2024b; Li et al., 2024).

## B    MATHEMATICAL BACKGROUND

We first expand on the properties mentioned within the main paper for Shapley values (Shapley, 1997) and SHAP (Lundberg & Lee, 2017) below.

### B.1    SHAPLEY VALUE PROPERTIES

**Property B.1 (Efficiency)**

$$\sum_{i \in N} \varphi_i(r) = r(N)$$

This property requires that the sum of the contributions across all players is equal to the model output.

**Property B.2 (Symmetry)** *If $r(S \cup i) = r(S \cup j)$   $\forall S \subseteq N \setminus \{i, j\}, \forall i, j \in N$ such that $i \neq j$, then:*

$$\varphi_i(r) = \varphi_j(r)$$

This property requires that two players that contribute equally will be assigned the same score.

**Property B.3 (Linearity)** *If we have two different reward functions $v$ and $w$:*

$$\varphi_i(r + t) = \varphi_i(r) + \varphi_i(t) \quad and \quad \varphi_i(ar) = a\varphi_i(r) \quad \forall a \in \mathbb{R}$$

The linearity property requires that the Shapley value of the sum of two (or more) reward functions is the same as the sum of their individual Shapley values.

**Property B.4 (Null player)** *If $r(S \cup \{i\}) = r(S)$   $\forall S \in N \setminus \{i\}$:*

$$\varphi_i(r) = 0$$

Finally, the Null player property requires that if a player does not contribute then they get assigned a score of 0.

## B.2 SHAP PROPERTIES

**Property B.5 (Local accuracy)**

$$f(x) = g(x') = \phi_0 + \sum_{i=1}^{M} (\phi_i \times x_i')$$

Local accuracy requires that if a model is approximated, then the output of the approximated model should match the original output.

**Property B.6 (Missingness)**

$$x_i' = 0 \implies \phi_i = 0$$

The Missingness property requires that if a feature is missing then it won't be given any attribution.

**Property B.7 (Consistency)** *Let $\mathbf{e}_i$ be the $M$ sized vector of 1s with a 0 at position $i$ and $\odot$ the Hadamard (elementwise) product. For any two models $f$ and $f'$ with attributions $\phi_i$ and $\phi_i'$ respectively, if:*

$$f'(h_x(z')) - f'(h_x(z' \odot \mathbf{e}_i)) \geq f(h_x(z')) - f(h_x(z' \odot \mathbf{e}_i)) \quad \forall z' \in \{0,1\}^M$$

*then $\phi_i' \geq \phi_i$*

Finally, the consistency property requires that if a feature's contribution stays constant or is increased due to a change in the model, then its attribution won't decrease.

**Definition B.1 (Additive feature attribution)** *An additive feature attribution method has explanation model:*

$$g(x') = \phi_0 + \sum_{i=1}^{M} (\phi_i \times x_i')$$

*where $x' \in \{0,1\}^M$, $M$ is the number of simplified features and $\phi_i \in \mathbb{R}$ is the attribution of each simplified feature.*

**Theorem B.1** *If we have explanation model $g$ defined as in Definition 3.2, then the only attribution satisfying Properties B.5 to B.7 is the Shapley value. In other words:*

$$\phi_i = \varphi_i(f)$$

From the SHAP (Lundberg & Lee, 2017) paper, only the Shapley value satisfies all of the above properties via Additive feature attribution.

## C IMPLEMENTATION DETAILS

FrozenBiLM and InternVideo were evaluated on a single 1080Ti GPU. VideoLLaMA2, LLaVA-Video, LongVA and VideoLLaMA3 were evaluated on a single GH200 GPU. All experiments took a total of $\sim 1800$ node hours/7200 GPU hours. For EgoSchema, MVBench and LVBench, frames are sampled uniformly from the single source videos (at the default framerate). However, as HD-EPIC takes multi-video input, we instead concatenate all videos from the VQA-tuple and then uniformly sample frames from this sequence. We pre-processed HD-EPIC into 1 FPS videos first.

In the name of reproducibility, these are the model checkpoints we used:

- FrozenBiLM (Yang et al., 2022) - CLIP ViT-L-14 and pre-trained on WebVid10M + How2QA

- InternVideo (Wang et al., 2022) - CLIP ViT-L-14 and pre-trained on MSRVTT
- VideoLLaMA2 (Cheng et al., 2024) - VideoLLaMA2-7B-16F
- LLaVA-Video (Zhang et al., 2024) - LLaVA-Video-7B-Qwen2
- LongVA (Zhang et al., 2025b) - LongVA-7B
- VideoLLaMA3 (Zhang et al., 2025a) - VideoLLaMA3-7B

As well as this, all code, dataset subsets, results and model checkpoints used in our experiments will be released after the review period to encourage reproducibility and usage of the framework.

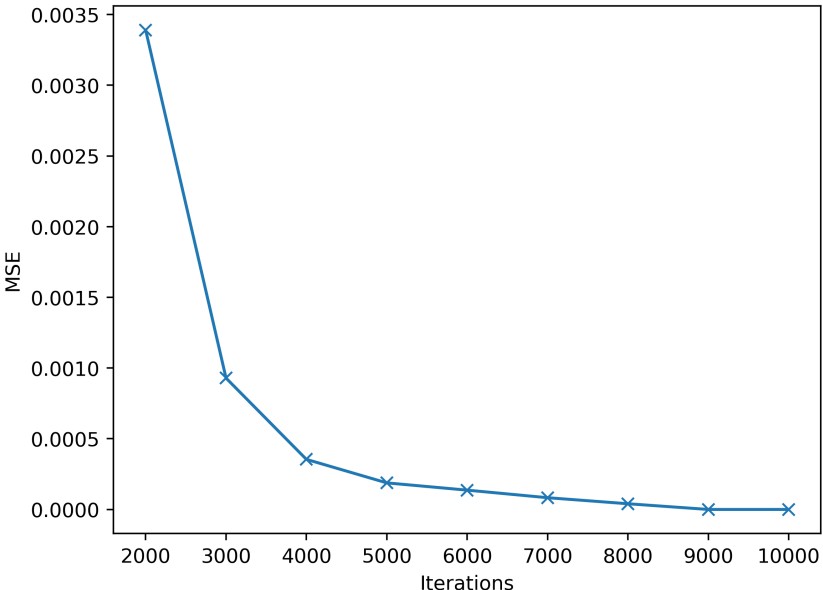

Figure 4: Plot of the MSE (mean squared error) of Shapley values at varying iterations for Frozen-BiLM on the longest EgoSchema question. Error is calculated against values for $10,000$ iterations.

To select $5000$ iterations as the value for our experiments, we ablated the parameter by varying it and selecting the point with a sensible error trade-off. In fig. 4 we have the results of the ablation and can see that $5000$ iterations is past the elbow of the curve and where the gradient of the error begins to flatten. As such, we selected the value for all experiments.

## D    SUBSET STATISTICS

Table 3: Statistics of the subsets used for evaluation.

| Dataset | # VQA-tuples | Avg. Video Length | Avg. Question Length | Avg. Answer Length |
| --- | --- | --- | --- | --- |
| EgoSchema | 50 | 180.00s | 24.36 words | 108.66 words |
| HD-EPIC | 60 | 1180.84s | 21.17 words | 56.75 words |
| MVBench | 60 | 16.09s | 13.32 words | 15.25 words |
| LVBench | 60 | 3800.13s | 10.25 words | 28.80 words |

We provide additional details and statistics of the subsets we used for the calculation of the Shapley values and the evaluation. Table 3, contains the number of VQA-tuples, as well as the average video length (of all unique videos corresponding to the subset), question length, and answer length.

# E QUANTITATIVE RESULTS

We provide additional results of experiments in the appendix. Firstly, we demonstrate that we can calculate Shapley values for open-ended VQA in appendix E.1. Secondly, we compare Shapley based rankings of frames to those generated by Gemini in appendix E.2. Then, we investigate the Modality Contribution and Per-Feature Contribution metrics for false logits in appendix E.3. After this, we showcase how masking *negatively* contributing features affects the results in appendix E.4, and the same for *positively* contributing features in appendix E.5. We visualise how video contributions vary across video context length in appendix E.6, and plot the distribution of Shapley values across all methods and datasets to compare how they differ in appendix E.7 Finally, we demonstrate how injecting new answers affects the masked performance in appendix E.8.

## E.1 OPEN-ENDED VISUAL QUESTION ANSWERING

Table 4: MC and PFC for each modality in the VQA-tuple. Calculated based on the Shapley values for the ground truth text for open-ended EgoSchema and averaged across all VQA-tuples. Here **blue** scores relate to large magnitudes of Shapley values, regardless of their sign, while **red** scores relate to values close to $0$.

|  | Modality Contribution | | Per-Feature Contribution | |
|---|---|---|---|---|
|  | V | Q | V | Q |
| L-V | 0.38 | 0.62 | 0.21 | 0.79 |
| VL3 | 0.69 | 0.31 | 0.25 | 0.75 |

Throughout the main paper we only discuss multiple-choice VQA because of the fact that its evaluation is simpler and it has a greater potential for textual bias. However, to demonstrate that our approach extends trivially to open-ended VQA, we include an example. In the open-ended scenario, the language model predicts a set of several output tokens, which then need to be compared to a piece of ground truth text. There is no longer a single logit to use as the reward function (and the set of logits of the generated tokens cannot be easily used because it is of variable length), so we decided to employ a captioning metric. We used ROUGE-L (Lin, 2004) (between the predicted and ground truth text) as the captioning metric. In table 4, we have the feature contributions for LLaVA-Video and VideoLLaMA3 on an open-ended version of EgoSchema where the answers have been removed from the input. We see that the ratio between video and question is extremely similar to that of the multiple-choice results and that the Per-Feature Contribution of video remains significantly smaller than the textual modality.

## E.2 GEMINI RANKING CORRELATION

```
You will be given frames from a video and a question with
multiple-choice answer options.
frame_0: {frame 0} ,..., frame_n-1:{frame n-1}
Question: {question}
Options: {answer choices}
You do not need to answer the question; return a comma
separated list of frame ids in the order of their importance
for answering the above question. Order the frame ids by
importance, do not leave them in chronological order. Respond
with exactly 180 frame ids, returning only this comma
separated list and excluding all other textual output.
```
Listing 1: Example system prompt for generating ranks for a VQA-tuple with 180 input frames.

To determine how much the framewise Shapley value contributions relate to common sense understanding of video, we compare them to a baseline generated from Gemini queries. In particular, we first ask Gemini to rank all of the input frames that would be sampled for a given model and dataset pair. The system prompt for querying Gemini is shown in listing 1. Then we calculate

the Spearman's correlation between this ranking and the ranking obtained by sorting the frames by absolute Shapley value. Plotting the correlation for each VQA-tuple in fig. 5, we see their distributions for several model/dataset combinations. As the modal correlation is always very close to $0$, there is little correlation between the two ranking systems, which aligns with our observations that the top 16 most influential frames often disagree with common sense reasoning. The models are often focusing on frames that are irrelevant for answering the question.

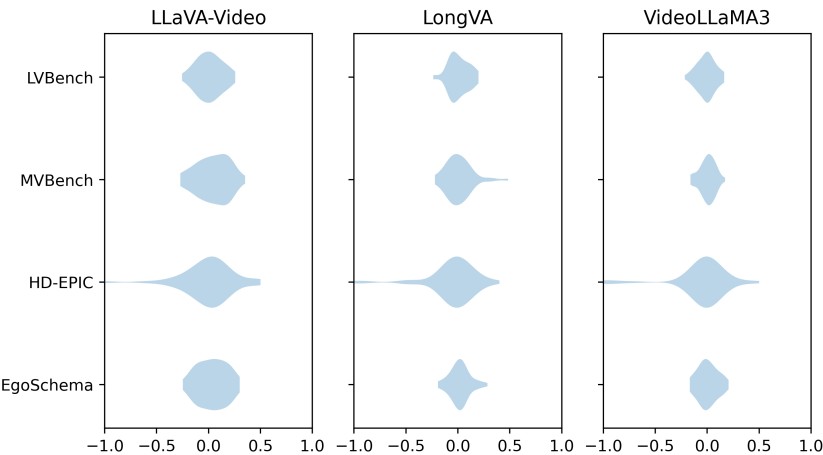

Figure 5: Violin plots of the Spearman's correlations between the Gemini rankings and the Shapley value rankings for the frames of VQA-tuple questions.

### E.3 CONTRIBUTIONS FOR FALSE LOGITS

Table 5: MC and PFC for each modality in the VQA-tuple. Calculated based on the Shapley values for the *false logits* averaged across all VQA-tuples. Here **blue** scores relate to large magnitudes of Shapley values, regardless of their sign, while **red** scores relate to values close to $0$.

(a) EgoSchema

| | Modality Contribution | | | Per-Feature Contribution | | | Acc |
|---|---|---|---|---|---|---|---|
| | V | Q | A | V | Q | A | |
| FBLM | 0.08 | 0.27 | 0.66 | 0.30 | 0.41 | 0.29 | 0.20 |
| IV | 0.22 | 0.37 | 0.41 | 0.54 | 0.35 | 0.11 | 0.36 |
| VL2 | 0.11 | 0.20 | 0.69 | 0.30 | 0.35 | 0.34 | 0.56 |
| L-V | 0.17 | 0.17 | 0.66 | 0.17 | 0.40 | 0.43 | 0.72 |
| LVA | 0.18 | 0.16 | 0.66 | 0.10 | 0.43 | 0.48 | 0.48 |
| VL3 | 0.37 | 0.14 | 0.49 | 0.17 | 0.43 | 0.40 | 0.70 |

(b) HD-EPIC

| | Modality Contribution | | | Per-Feature Contribution | | | Acc |
|---|---|---|---|---|---|---|---|
| | V | Q | A | V | Q | A | |
| FBLM | 0.09 | 0.54 | 0.38 | 0.20 | 0.55 | 0.25 | 0.22 |
| IV | 0.33 | 0.50 | 0.17 | 0.53 | 0.39 | 0.08 | 0.15 |
| VL2 | 0.19 | 0.28 | 0.53 | 0.29 | 0.29 | 0.41 | 0.28 |
| L-V | 0.22 | 0.29 | 0.49 | 0.19 | 0.36 | 0.45 | 0.35 |
| LVA | 0.22 | 0.23 | 0.55 | 0.14 | 0.32 | 0.55 | 0.35 |
| VL3 | 0.41 | 0.22 | 0.37 | 0.19 | 0.40 | 0.42 | 0.35 |

(c) MVBench

| | Modality Contribution | | | Per-Feature Contribution | | | Acc |
|---|---|---|---|---|---|---|---|
| | V | Q | A | V | Q | A | |
| FBLM | 0.11 | 0.52 | 0.37 | 0.13 | 0.50 | 0.36 | 0.40 |
| IV | 0.27 | 0.52 | 0.21 | 0.33 | 0.49 | 0.18 | 0.42 |
| VL2 | 0.23 | 0.28 | 0.50 | 0.20 | 0.28 | 0.52 | 0.62 |
| L-V | 0.25 | 0.28 | 0.47 | 0.07 | 0.34 | 0.59 | 0.65 |
| LVA | 0.18 | 0.28 | 0.54 | 0.02 | 0.33 | 0.64 | 0.45 |
| VL3 | 0.44 | 0.26 | 0.30 | 0.06 | 0.44 | 0.51 | 0.65 |

(d) LVBench

| | Modality Contribution | | | Per-Feature Contribution | | | Acc |
|---|---|---|---|---|---|---|---|
| | V | Q | A | V | Q | A | |
| FBLM | 0.15 | 0.45 | 0.40 | 0.21 | 0.47 | 0.32 | 0.30 |
| IV | 0.34 | 0.44 | 0.22 | 0.42 | 0.44 | 0.14 | 0.25 |
| VL2 | 0.26 | 0.27 | 0.47 | 0.25 | 0.32 | 0.43 | 0.27 |
| L-V | 0.30 | 0.27 | 0.43 | 0.10 | 0.40 | 0.49 | 0.42 |
| LVA | 0.39 | 0.19 | 0.42 | 0.07 | 0.35 | 0.58 | 0.35 |
| VL3 | 0.54 | 0.17 | 0.29 | 0.10 | 0.41 | 0.49 | 0.48 |

Within the main paper we focused on contributions based on ground truth logits in section 4.2, whereas here in the appendix we provide an exploration into how contributions differ based on the false logits. As there are several false logits, instead of a single ground truth logit, we average the Shapley values across the false logits. Table 5, highlights these results. Overall, we see similar trends between the two tables across all datasets and models: namely that video is important as an entire

modality for VideoLLaMA3, but that per-frame contributions remain low for long context models. As well as this, the question remains under-represented compared to the answers.

### E.4 MASKING NEGATIVE CONTRIBUTIONS

Table 6: *How does masking the input based upon **negative** Per-Feature Contributions affect accuracy?* We mask all negative features across the entire input (joint) or each modality separately. "None" represents the vanilla accuracy. Here green/red refers to an increase/decrease in accuracy.

| Model | Masking | EgoSchema | HD-EPIC | MVBench | LVBench |
|---|---|---|---|---|---|
| FrozenBiLM | None | 0.20 | 0.22 | 0.40 | 0.30 |
| | All | +0.14 | +0.02 | +0.10 | +0.00 |
| | Video | +0.06 | +0.03 | +0.10 | +0.10 |
| | Question | +0.00 | +0.07 | +0.08 | +0.03 |
| | Answer | +0.14 | +0.02 | +0.10 | +0.00 |
| InternVideo | None | 0.36 | 0.15 | 0.42 | 0.25 |
| | All | +0.10 | -0.03 | -0.03 | +0.05 |
| | Video | +0.04 | +0.08 | +0.02 | +0.08 |
| | Question | +0.08 | +0.00 | +0.05 | +0.07 |
| | Answer | +0.10 | -0.03 | -0.03 | +0.05 |
| VideoLLaMA2 | None | 0.56 | 0.28 | 0.62 | 0.27 |
| | All | +0.10 | +0.08 | +0.05 | +0.03 |
| | Video | +0.02 | +0.07 | -0.02 | +0.22 |
| | Question | +0.06 | +0.07 | +0.07 | +0.12 |
| | Answer | +0.10 | +0.08 | +0.05 | +0.03 |
| LLaVA-Video | None | 0.72 | 0.35 | 0.65 | 0.42 |
| | All | +0.08 | +0.03 | +0.08 | +0.02 |
| | Video | +0.06 | +0.05 | +0.07 | +0.20 |
| | Question | +0.08 | +0.03 | +0.13 | +0.02 |
| | Answer | +0.08 | +0.03 | +0.08 | +0.02 |
| LongVA | None | 0.48 | 0.35 | 0.45 | 0.35 |
| | All | +0.18 | +0.12 | +0.10 | +0.03 |
| | Video | +0.10 | +0.02 | +0.00 | +0.17 |
| | Question | +0.16 | -0.02 | +0.10 | +0.13 |
| | Answer | +0.18 | +0.12 | +0.10 | +0.03 |
| VideoLLaMA3 | None | 0.70 | 0.35 | 0.65 | 0.48 |
| | All | +0.10 | +0.03 | +0.05 | -0.02 |
| | Video | +0.12 | +0.15 | +0.07 | +0.12 |
| | Question | +0.02 | +0.12 | +0.07 | +0.10 |
| | Answer | +0.10 | +0.03 | +0.05 | -0.02 |

In the main paper in section 4.3 we demonstrated the effect upon performance when masking entire modalities. Now, in table 6 we mask individual features if their Shapley values are negative. For the sake of fairness, only the ground truth answer can be masked to ensure that the masking does not just greedily remove all of the false answer text (thus trivialising the multiple-choice). The intuition here is to verify the extent to which the Shapley values help inform the potential accuracy ceiling of the model. Generally, all models gain performance when distractors are masked. For EgoSchema, HD-EPIC and MVBench, the biggest increases tend to be answer, question, then video. In LVBench, masking video becomes more important, likely because the context is always long, and the question may only be relevant to a small portion of frames. VideoLLaMA3 gains a significant performance boost from masking video in EgoSchema, HD-EPIC and LVBench but requires masking (on average) 35%, 39% and 40% of frames respectively, compared to 28% for MVBench which only gains 7% accuracy. To exploit the video modality to its best ability, even the strongest model we test needs to remove more than a third of the frames. This points toward better frame sampling methods to be an important consideration for future models. Overall, we see a tendency for accuracy to be explained well by the Shapley values.

## E.5 MASKING POSITIVE CONTRIBUTIONS

Table 7: *How does masking the input based upon **positive** Per-Feature Contributions affect accuracy?* We mask all negative features across the entire input (joint) or each modality separately. "None" represents the vanilla accuracy. Here **green**/**red** refers to an increase/decrease in accuracy.

| Model | Masking | EgoSchema | HD-EPIC | MVBench | LVBench |
|---|---|---|---|---|---|
| FrozenBiLM | None | 0.20 | 0.22 | 0.40 | 0.30 |
| | All | +0.14 | +0.03 | +0.10 | +0.00 |
| | Video | -0.02 | -0.02 | -0.08 | -0.07 |
| | Question | +0.04 | +0.02 | -0.13 | +0.00 |
| | Answer | +0.14 | +0.03 | +0.10 | +0.00 |
| InternVideo | None | 0.36 | 0.15 | 0.42 | 0.25 |
| | All | +0.10 | -0.03 | -0.03 | +0.05 |
| | Video | +0.02 | +0.00 | -0.03 | +0.00 |
| | Question | +0.00 | +0.05 | +0.02 | -0.03 |
| | Answer | +0.10 | -0.03 | -0.03 | +0.05 |
| VideoLLaMA2 | None | 0.56 | 0.28 | 0.62 | 0.27 |
| | All | +0.06 | +0.02 | +0.00 | +0.05 |
| | Video | +0.00 | -0.10 | -0.10 | -0.08 |
| | Question | -0.02 | -0.12 | -0.20 | -0.07 |
| | Answer | +0.06 | +0.02 | +0.00 | +0.05 |
| LLaVA-Video | None | 0.72 | 0.35 | 0.65 | 0.42 |
| | All | +0.02 | -0.07 | +0.07 | +0.00 |
| | Video | -0.26 | -0.17 | -0.10 | -0.20 |
| | Question | -0.02 | -0.17 | -0.20 | -0.13 |
| | Answer | +0.02 | -0.07 | +0.07 | +0.00 |
| LongVA | None | 0.48 | 0.35 | 0.45 | 0.35 |
| | All | +0.04 | -0.02 | +0.00 | -0.07 |
| | Video | -0.12 | -0.10 | -0.07 | -0.12 |
| | Question | -0.22 | -0.23 | -0.17 | -0.05 |
| | Answer | +0.04 | -0.02 | +0.00 | -0.07 |
| VideoLLaMA3 | None | 0.70 | 0.35 | 0.65 | 0.48 |
| | All | +0.02 | +0.05 | +0.05 | -0.10 |
| | Video | -0.06 | -0.08 | -0.02 | -0.12 |
| | Question | -0.02 | -0.13 | -0.15 | -0.15 |
| | Answer | +0.02 | +0.05 | +0.05 | -0.10 |

Here we instead mask individual features if their Shapley values are positive, to see how the accuracy is affected if *all* features are distractors. Again, for the sake of fairness, only the ground truth answer can be masked. Table 7 shows the results. Compared to the results in the main paper, we see that masking the video or question decreases the accuracy significantly, confirming that features contributing positively are required for strong performance. However, masking the answers actually results in small improvements in accuracy, likely because the masking of the ground truth answer makes this option stand out and allows the language model to exploit the difference.

## E.6 VIDEO CONTRIBUTION VS. VIDEO CONTEXT

We visualise the relationship between video Per-Feature Contribution ($PFC_V$) and video context length in fig. 6. HD-EPIC was used for this because it contains videos with the highest variance in context length. The Pearson correlation between these two variables is $-0.36$, suggesting a slight negative correlation, which means that as the video input increases in length, the less each frame contributes to the output.

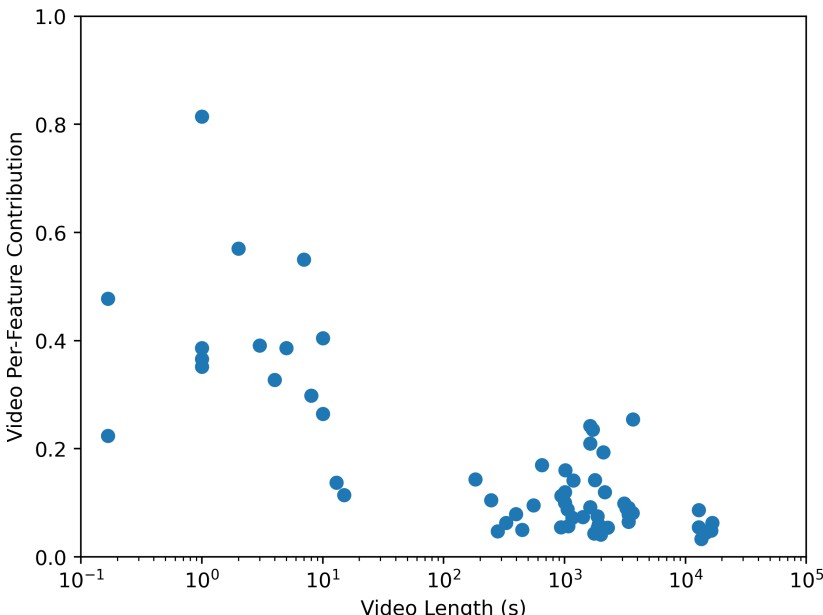

Figure 6: PFC$_V$ plotted against video context length in seconds for the HD-EPIC subset we used. The x-axis is log base 10 scale.

### E.7 DISTRIBUTIONS OF SHAPLEY VALUES

We showcase the per modality distributions of Shapley values across all models and datasets as violin plots in fig. 7. We plot a violin of the set of Shapley values for all of the video, question and answer features separately for a given dataset subset. As we move down the plots, we see that the height of the violin for video decreases significantly, and that it increases for the answers. On the other hand, the violins for the question presents similar distributions throughout. MVBench and LVBench demonstrate larger answer contributions than EgoSchema and HD-EPIC, further indicating that these datasets are skewing attention towards discriminating between multiple-choice answers. Overall, we see that the larger models utilising LLMs as backbones tend to be biased towards the text modalities.

### E.8 ANSWER REPLACEMENT MASKING

In this subsection we plot the accuracy of models (similarly to section 4.3) when masking modalities for the datasets with injected negatives from answer replacement. We see in table 8 that "Easy" answer replacement significantly improves performance in all scenarios as it becomes more trivial for the model to discern the true negative. Then, in table 9, table 10, table 11 and table 12 we have the performance on each of the "New" replacement types. In general, we see that adding these extra negatives often increases the accuracy lost when masking out video, particularly for EgoSchema and LVBench. Masking out the question can also have an increased effect on performance, especially for LVBench. The differences are significantly less pronounced for HD-EPIC, likely because the unmasked performance drops as low as $10\%$. To summarise, these tables show support for our findings in section 4.4 that injecting new negatives can have an effect on the contributions of the video and question modalities.

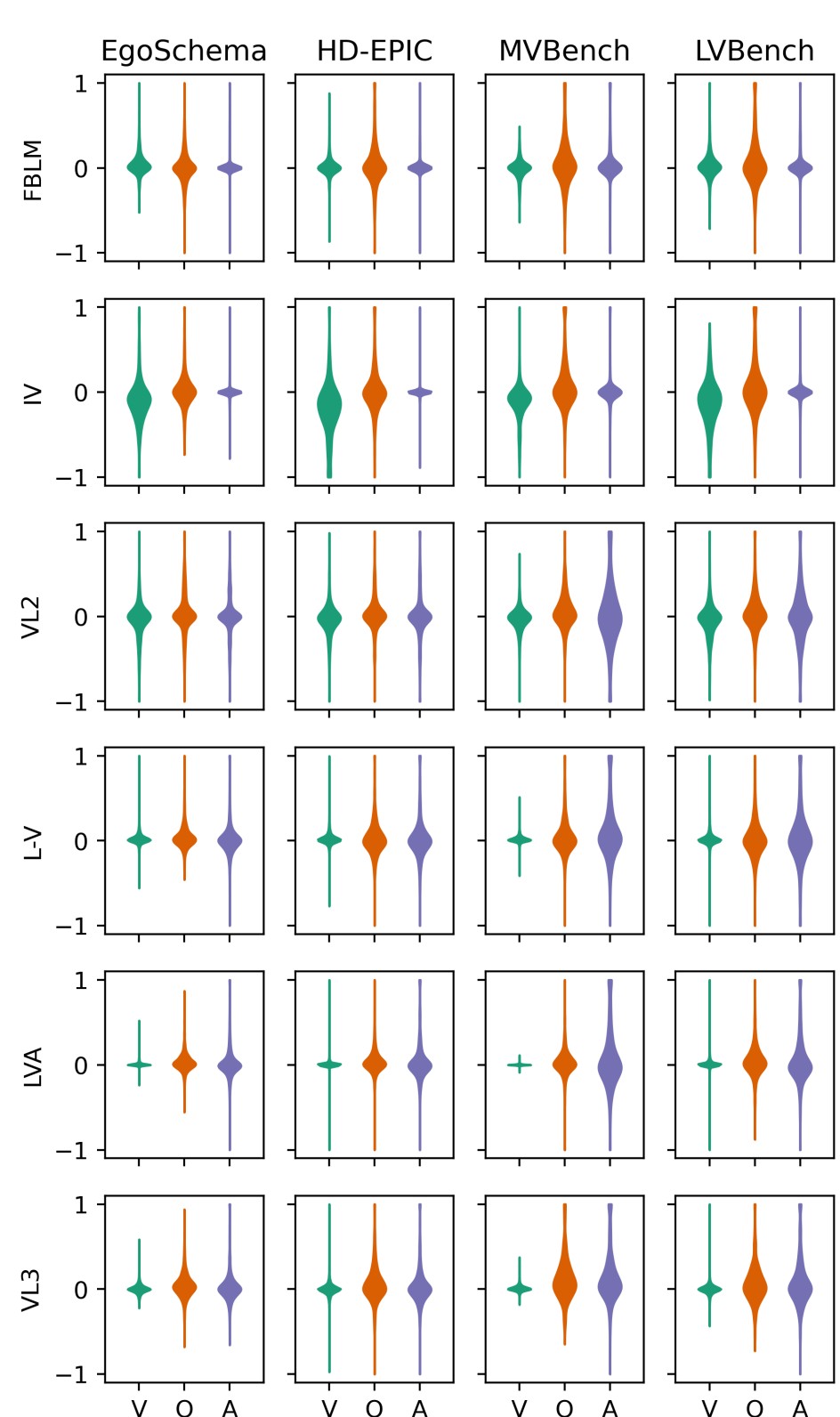

Figure 7: Violin plots of the Shapley values per modality across all models and datasets, for the ground truth logits.

Table 8: Modality masking performance for "Easy" answer replacement. We mask either all input features ("All"), or each modality separately and compare to baseline performance. "None" represents the vanilla accuracy. Here **green**/**red** refers to an increase/decrease in accuracy compared to baseline.

| Model | Masking | EgoSchema | HD-EPIC | LVBench |
|---|---|---|---|---|
| VideoLLaMA2 | None | 0.94 | 0.67 | 0.83 |
| | All | -0.68 | -0.25 | -0.58 |
| | Video | -0.40 | -0.03 | -0.10 |
| | Question | -0.24 | -0.47 | -0.35 |
| | Answer | -0.68 | -0.25 | -0.58 |
| LLaVA-Video | None | 0.98 | 0.77 | 0.97 |
| | All | -0.80 | -0.35 | -0.60 |
| | Video | -0.36 | 0.02 | -0.15 |
| | Question | -0.04 | -0.42 | -0.30 |
| | Answer | -0.80 | -0.35 | -0.60 |
| LongVA | None | 0.88 | 0.60 | 0.88 |
| | All | -0.70 | -0.27 | -0.45 |
| | Video | -0.34 | -0.03 | -0.05 |
| | Question | -0.14 | -0.33 | -0.17 |
| | Answer | -0.70 | -0.27 | -0.45 |
| VideoLLaMA3 | None | 1.00 | 0.77 | 0.95 |
| | All | -0.82 | -0.52 | -0.58 |
| | Video | -0.40 | -0.02 | -0.03 |
| | Question | -0.10 | -0.45 | -0.22 |
| | Answer | -0.82 | -0.52 | -0.58 |

Table 9: Modality masking performance for "New-5" answer replacement. We mask either all input features ("All"), or each modality separately and compare to baseline performance. "None" represents the vanilla accuracy. Here **green**/**red** refers to an increase/decrease in accuracy compared to baseline.

| Model | Masking | EgoSchema | HD-EPIC | LVBench |
|---|---|---|---|---|
| VideoLLaMA2 | None | 0.48 | 0.18 | 0.37 |
| | All | -0.48 | -0.15 | -0.35 |
| | Video | -0.28 | -0.10 | -0.10 |
| | Question | 0.02 | -0.05 | -0.20 |
| | Answer | -0.48 | -0.15 | -0.35 |
| LLaVA-Video | None | 0.66 | 0.10 | 0.38 |
| | All | -0.56 | -0.03 | -0.32 |
| | Video | -0.40 | 0.02 | -0.22 |
| | Question | -0.04 | -0.02 | -0.18 |
| | Answer | -0.56 | -0.03 | -0.32 |
| LongVA | None | 0.44 | 0.15 | 0.30 |
| | All | -0.38 | -0.10 | -0.25 |
| | Video | -0.22 | -0.07 | -0.15 |
| | Question | -0.12 | -0.08 | -0.07 |
| | Answer | -0.38 | -0.10 | -0.25 |
| VideoLLaMA3 | None | 0.74 | 0.15 | 0.30 |
| | All | -0.74 | -0.07 | -0.27 |
| | Video | -0.58 | -0.05 | -0.03 |
| | Question | -0.08 | -0.02 | -0.13 |
| | Answer | -0.74 | -0.07 | -0.27 |

Table 10: Modality masking performance for "New-10" answer replacement. We mask either all input features ("All"), or each modality separately and compare to baseline performance. "None" represents the vanilla accuracy. Here green/red refers to an increase/decrease in accuracy compared to baseline.

| Model | Masking | EgoSchema | HD-EPIC | LVBench |
|---|---|---|---|---|
| VideoLLaMA2 | None | 0.48 | 0.10 | 0.25 |
| | All | -0.44 | -0.05 | -0.23 |
| | Video | -0.26 | 0.00 | -0.05 |
| | Question | -0.06 | 0.05 | -0.10 |
| | Answer | -0.44 | -0.05 | -0.23 |
| LLaVA-Video | None | 0.68 | 0.12 | 0.32 |
| | All | -0.62 | -0.08 | -0.20 |
| | Video | -0.48 | 0.02 | -0.17 |
| | Question | -0.08 | -0.05 | -0.10 |
| | Answer | -0.62 | -0.08 | -0.20 |
| LongVA | None | 0.44 | 0.10 | 0.33 |
| | All | -0.38 | -0.03 | -0.28 |
| | Video | -0.30 | -0.02 | -0.15 |
| | Question | -0.16 | -0.05 | -0.10 |
| | Answer | -0.38 | -0.03 | -0.28 |
| VideoLLaMA3 | None | 0.68 | 0.10 | 0.40 |
| | All | -0.58 | -0.05 | -0.37 |
| | Video | -0.40 | 0.02 | -0.15 |
| | Question | -0.06 | -0.03 | -0.18 |
| | Answer | -0.58 | -0.05 | -0.37 |

Table 11: Modality masking performance for "New-15" answer replacement. We mask either all input features ("All"), or each modality separately and compare to baseline performance. "None" represents the vanilla accuracy. Here green/red refers to an increase/decrease in accuracy compared to baseline.

| Model | Masking | EgoSchema | HD-EPIC | LVBench |
|---|---|---|---|---|
| VideoLLaMA2 | None | 0.42 | 0.17 | 0.23 |
| | All | -0.40 | -0.10 | -0.18 |
| | Video | -0.18 | -0.05 | -0.05 |
| | Question | 0.02 | -0.03 | -0.07 |
| | Answer | -0.40 | -0.10 | -0.18 |
| LLaVA-Video | None | 0.66 | 0.15 | 0.32 |
| | All | -0.54 | -0.10 | -0.28 |
| | Video | -0.34 | -0.07 | -0.13 |
| | Question | 0.00 | -0.05 | -0.08 |
| | Answer | -0.54 | -0.10 | -0.28 |
| LongVA | None | 0.34 | 0.13 | 0.22 |
| | All | -0.26 | 0.02 | -0.17 |
| | Video | -0.22 | 0.03 | -0.07 |
| | Question | -0.04 | -0.08 | 0.02 |
| | Answer | -0.26 | 0.02 | -0.17 |
| VideoLLaMA3 | None | 0.66 | 0.15 | 0.37 |
| | All | -0.56 | -0.08 | -0.25 |
| | Video | -0.34 | 0.00 | -0.13 |
| | Question | -0.04 | -0.03 | -0.15 |
| | Answer | -0.56 | -0.08 | -0.25 |

Table 12: Modality masking performance for "New-20" answer replacement. We mask either all input features ("All"), or each modality separately and compare to baseline performance. "None" represents the vanilla accuracy. Here green/red refers to an increase/decrease in accuracy compared to baseline.

| Model | Masking | EgoSchema | HD-EPIC | LVBench |
|---|---|---|---|---|
| VideoLLaMA2 | None | 0.48 | 0.27 | 0.27 |
| | All | -0.38 | -0.15 | -0.18 |
| | Video | -0.12 | -0.10 | 0.00 |
| | Question | 0.08 | -0.07 | -0.07 |
| | Answer | -0.38 | -0.15 | -0.18 |
| LLaVA-Video | None | 0.68 | 0.17 | 0.38 |
| | All | -0.60 | -0.07 | -0.28 |
| | Video | -0.48 | -0.08 | -0.15 |
| | Question | 0.02 | -0.05 | -0.08 |
| | Answer | -0.60 | -0.07 | -0.28 |
| LongVA | None | 0.40 | 0.20 | 0.25 |
| | All | -0.32 | 0.00 | -0.18 |
| | Video | -0.24 | -0.03 | -0.07 |
| | Question | -0.04 | -0.07 | 0.00 |
| | Answer | -0.32 | 0.00 | -0.18 |
| VideoLLaMA3 | None | 0.74 | 0.15 | 0.45 |
| | All | -0.64 | -0.05 | -0.35 |
| | Video | -0.48 | 0.03 | -0.18 |
| | Question | -0.08 | -0.02 | -0.15 |
| | Answer | -0.64 | -0.05 | -0.35 |

## F    QUALITATIVE RESULTS

Here, we provide further qualitative results of our experimental results. First, in appendix F.1, we showcase all remaining heatmaps across all datasets. Next, we highlight further examples of entire VQA-tuple visualisations in appendix F.2. Finally, we also provide wordclouds showcasing Shapley values of specific words in the subsets in appendix F.3.

### F.1    HEATMAPS OF SHAPLEY VALUES

Expanding from the example in the main paper for VideoLLaMA3, we present the heatmaps across all other methods and datasets. FrozenBiLM in fig. 8, InternVideo in fig. 9, VideoLLaMA2 in fig. 10, LLaVA-Video in fig. 11, and LongVA can be found in fig. 12. Note that we truncate these figures for readability in a similar fashion for the heatmaps in the main paper. We find similar trends across all models in which there is a clear separation between videos and question/answer values. However, for VideoLLaMA2 and InternVideo we find that the video frames are distinguished often via a negative Shapley value, rather than a lower magnitude. Similarly to the results in the main paper for VideoLLaMA3, there is no strong temporal bias across the features (i.e. similar columns of values across the questions). For long context models, we again see that individual frames have low contributions.

### F.2    FURTHER QUALITATIVE EXAMPLES

We showcase further qualitative examples from VideoLLaMA3 across all datasets including EgoSchema in fig. 13, HD-EPIC in fig. 14, MVBench in fig. 15, and LVBench in fig. 16. We see similar trends to the example shown in the main paper in which VideoLLaMA3 will attribute negative scores to words in negative answers that match those in the ground truth answer (i.e. "paintbrush" in fig. 13). We also see a tendency for the sign of the frame contributions to not correspond with common sense. For example, in fig. 13, frames containing the mentioned "cup of water" are actually negatively attributed, which is strange considering it is the topic of the question. In fig. 15 the frame attributions appear to make more sense, as the model is positively influenced by early frames of the object at the start of motion and latter frames of the object at the end of motion, largely disregarding more intermediate frames. Although this is sensible, the question can essentially be solved solely by identifying the translation of the object between two frames, meaning it requires little temporal context. Therefore, these valid attributions are indicative of the simplicity of the underlying dataset. Finally, in fig. 14 we notice that none of the selected frames actually contain the ground truth object (the Jack of Spades card). Checking the sampled frames, we found that a frame containing this object is sampled in the input, but that its attribution is close to zero, providing more evidence that the model is not necessarily being guided by relevant frames. Overall, the video frames also have the same tendency to show lower peaks than the question/answer and the $PFC_V$ scores continue to be less than 0.1 across all the examples.

### F.3    WORD CLOUDS OF SHAPLEY VALUES

In this section, we explore how words are attributed based on their frequency within each of the dataset subsets. Specifically, we plot word clouds for each method on each dataset, combining all question and answers, so that the size of the word is proportional to its frequency. The colour of each word is calculated as the word's average Shapley value (for the ground truth logit) across all of its instances, with **blue** representing positively attributed words and **red** representing negatively attributed words.

The wordcloud for FrozenBiLM is in fig. 17, InternVideo is in fig. 18, VideoLLaMA2 is in fig. 19, LLaVA-Video is in fig. 20, LongVA is in fig. 21, and VideoLLaMA3 is in fig. 22. Stronger models, such as LLaVA-Video and VideoLLaMA3, tend to assign common words with a positive attribution, though sometimes "video" is seen as negative. Otherwise, we see dataset specific objects, actions, and adjectives get high attribution values. For example, "cylinder" and "cube" in MVBench and "left", "right", and "counter" in HD-EPIC. Apart from VideoLLaMA2, most words tend to have a positive attribution score, again showcasing the preference for the question and answer modalities over the video modality. In general, we fail to see any obvious biases towards specific types of words (verbs or nouns).

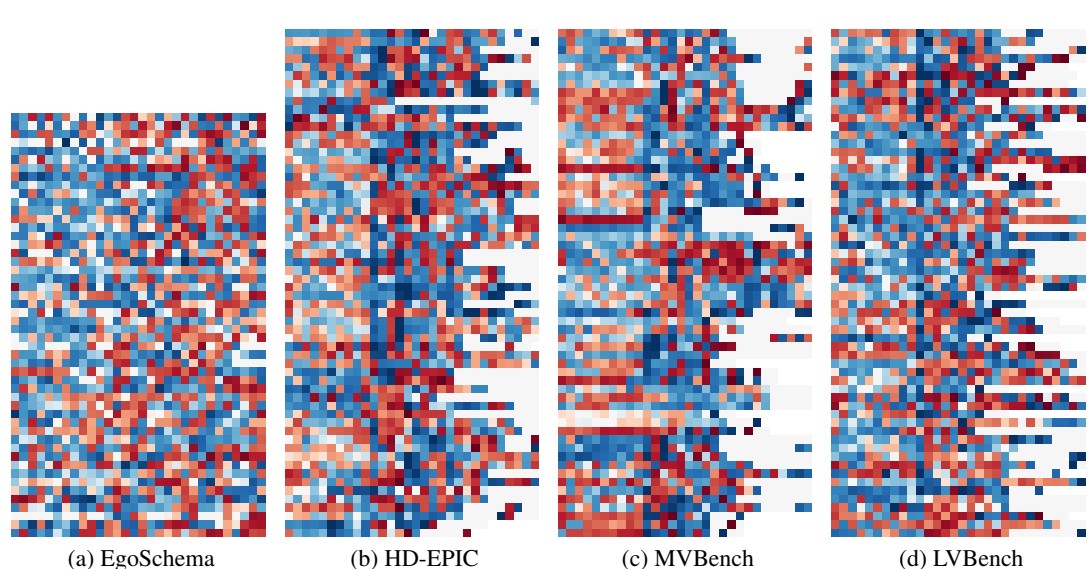

|     |     |     |     |
| --- | --- | --- | --- |
| (a) EgoSchema | (b) HD-EPIC | (c) MVBench | (d) LVBench |

Figure 8: Matrix of Shapley values per subset for FrozenBiLM, where each row, left-to-right, represents the features of a VQA-tuple. Rows are truncated to a maximum of 30 features with the first 10 values representing video frames.

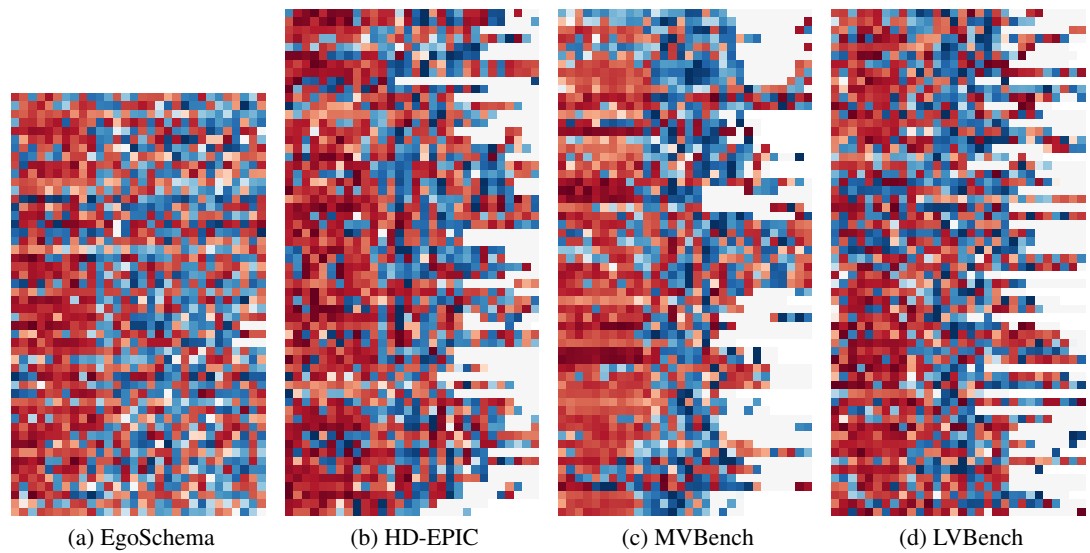

|     |     |     |     |
| --- | --- | --- | --- |
| (a) EgoSchema | (b) HD-EPIC | (c) MVBench | (d) LVBench |

Figure 9: Matrix of Shapley values per subset for InternVideo, where each row, left-to-right, represents the features of a VQA-tuple. Rows are truncated to a maximum of 30 features with the first 10 values representing video frames.

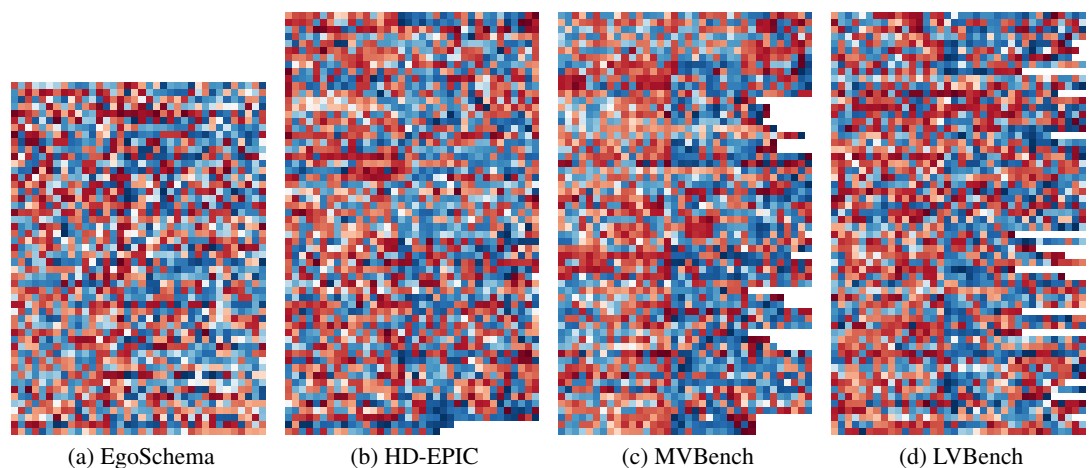

|  |  |  |  |
|---|---|---|---|
| (a) EgoSchema | (b) HD-EPIC | (c) MVBench | (d) LVBench |

Figure 10: Matrix of Shapley values per subset for VideoLLaMA2, where each row, left-to-right, represents the features of a VQA-tuple. Rows are truncated to a maximum of 36 features with the first 16 values representing video frames.

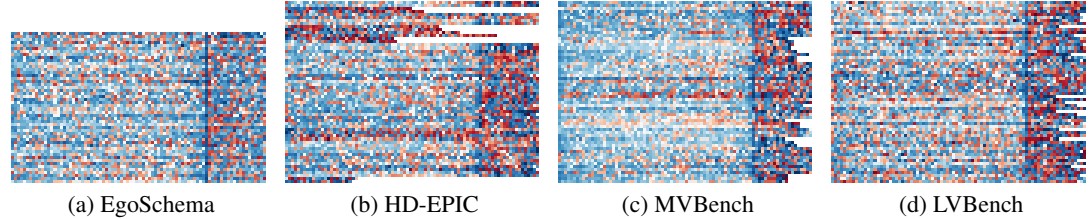

|  |  |  |  |
|---|---|---|---|
| (a) EgoSchema | (b) HD-EPIC | (c) MVBench | (d) LVBench |

Figure 11: Matrix of Shapley values per subset for LLaVA-Video, where each row, left-to-right, represents the features of a VQA-tuple. Rows are truncated to a maximum of 84 features with the first 64 values representing video frames.

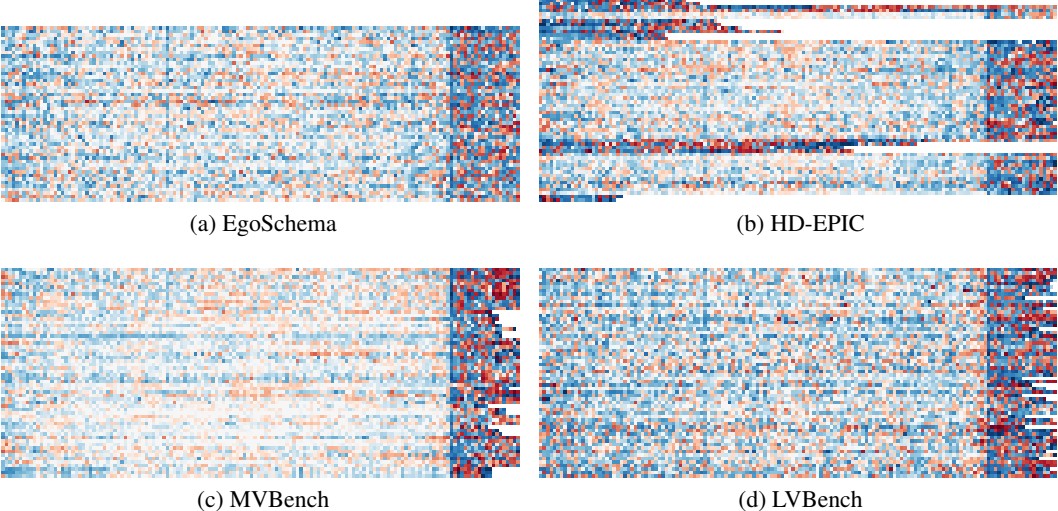

|  |  |
|---|---|
| (a) EgoSchema | (b) HD-EPIC |
| (c) MVBench | (d) LVBench |

Figure 12: Matrix of Shapley values per subset for LongVA, where each row, left-to-right, represents the features of a VQA-tuple. Rows are truncated to a maximum of 148 features with the first 128 values representing video frames.

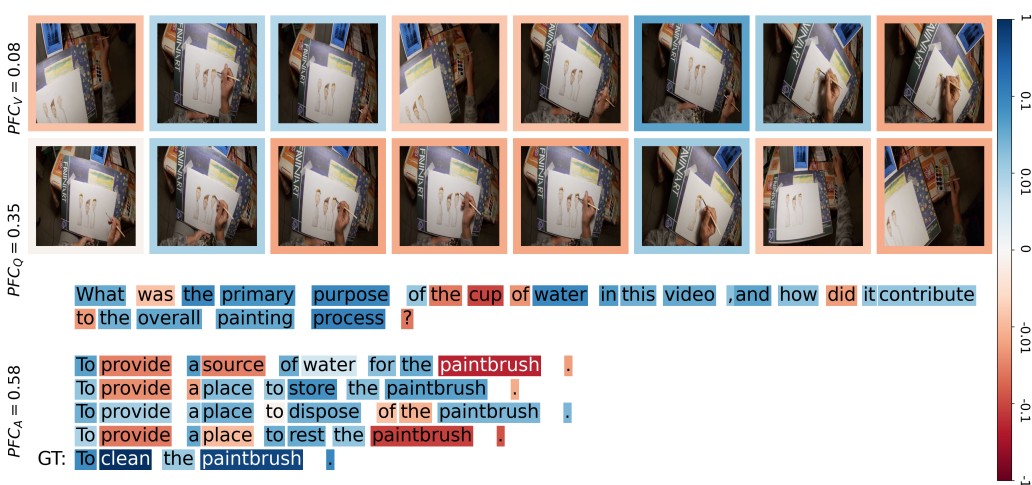

Figure 13: Qualitative figure of an example from EgoSchema evaluated using VideoLLaMA3. For brevity, we select the 16 most important frames, ranked by the magnitude of their Shapley values. Here **blue** represents positively attributed inputs whereas **red** represents negatively attributed inputs.

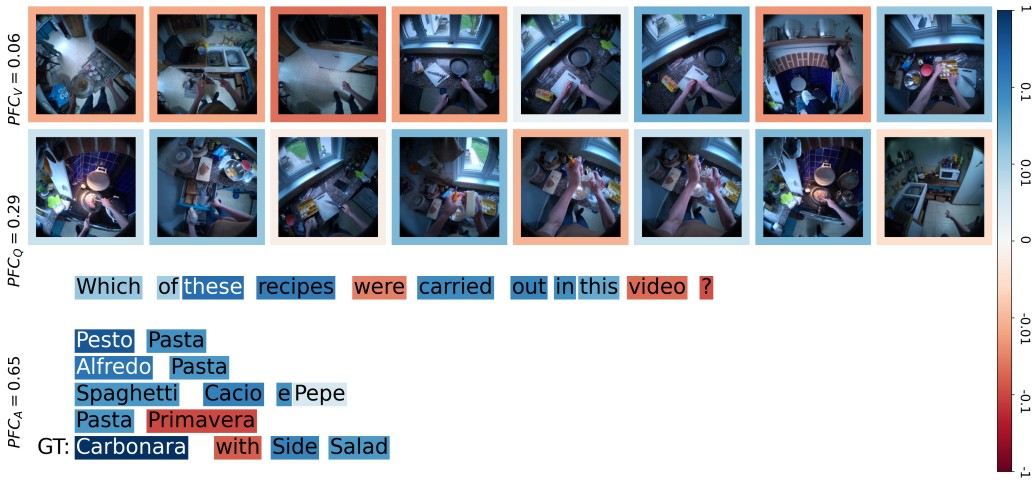

Figure 14: Qualitative figure of an example from HD-EPIC evaluated using VideoLLaMA3. For brevity, we select the 16 most important frames, ranked by the magnitude of their Shapley values. Here **blue** represents positively attributed inputs whereas **red** represents negatively attributed inputs.

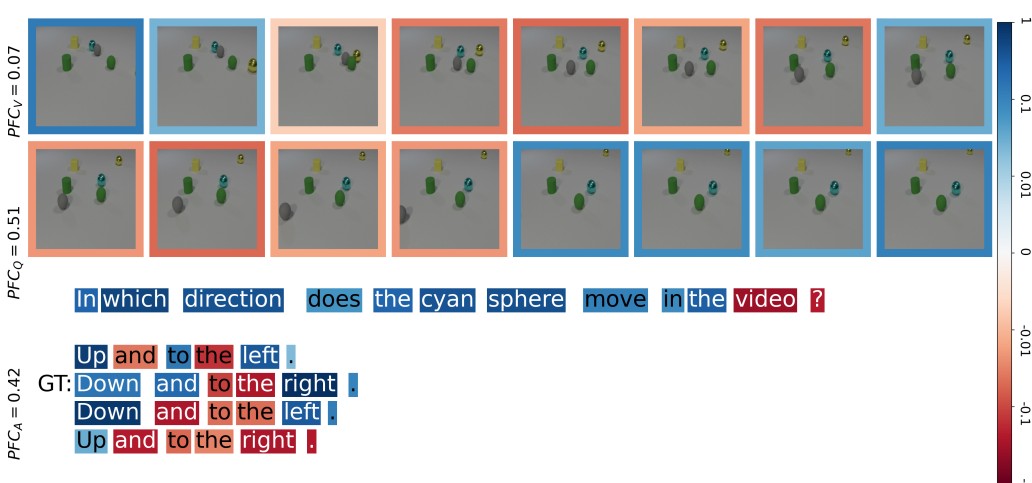

Figure 15: Qualitative figure of an example from MVBench evaluated using VideoLLaMA3. For brevity, we select the 16 most important frames, ranked by the magnitude of their Shapley values. Here **blue** represents positively attributed inputs whereas **red** represents negatively attributed inputs.

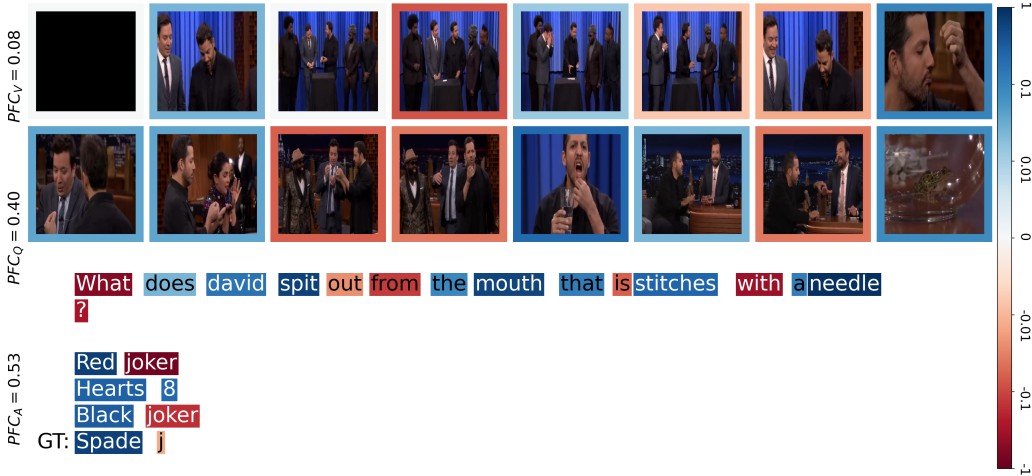

Figure 16: Qualitative figure of an example from LVBench evaluated using VideoLLaMA3. For brevity, we select the 16 most important frames, ranked by the magnitude of their Shapley values. Here **blue** represents positively attributed inputs whereas **red** represents negatively attributed inputs.

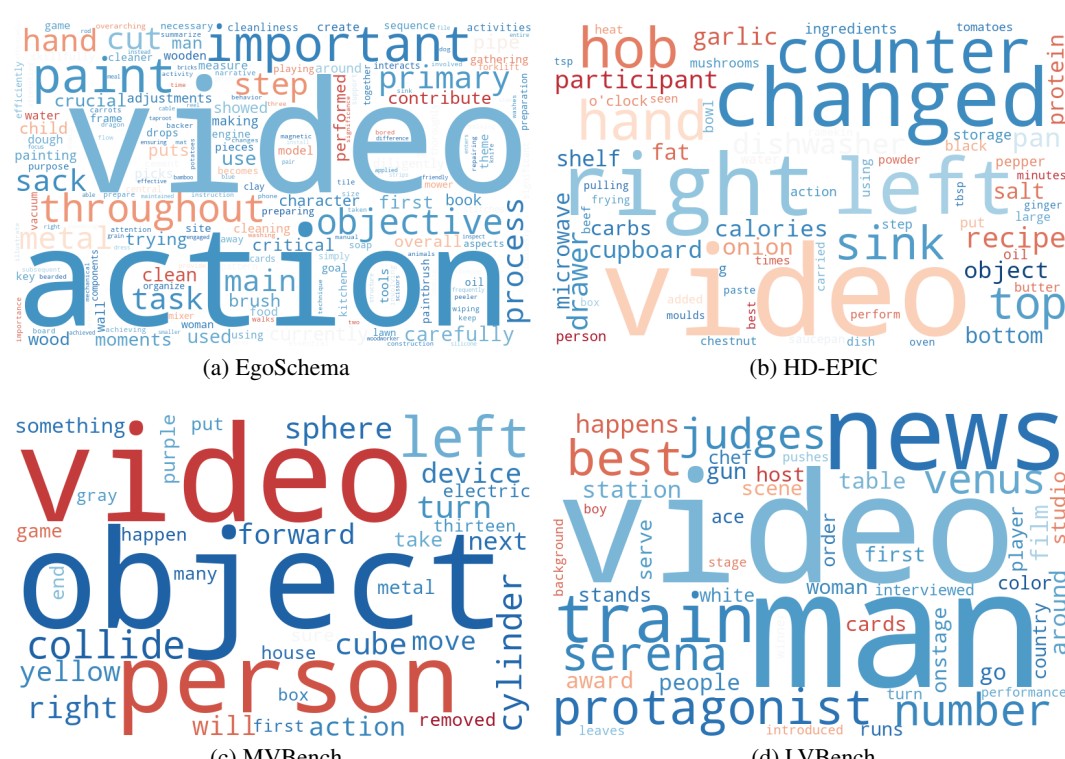

Figure 17: FrozenBiLM word clouds showing the frequency of the words within the dataset subset via their size and the attribution by their colour, **blue** for positive attribution and **red** for negative.

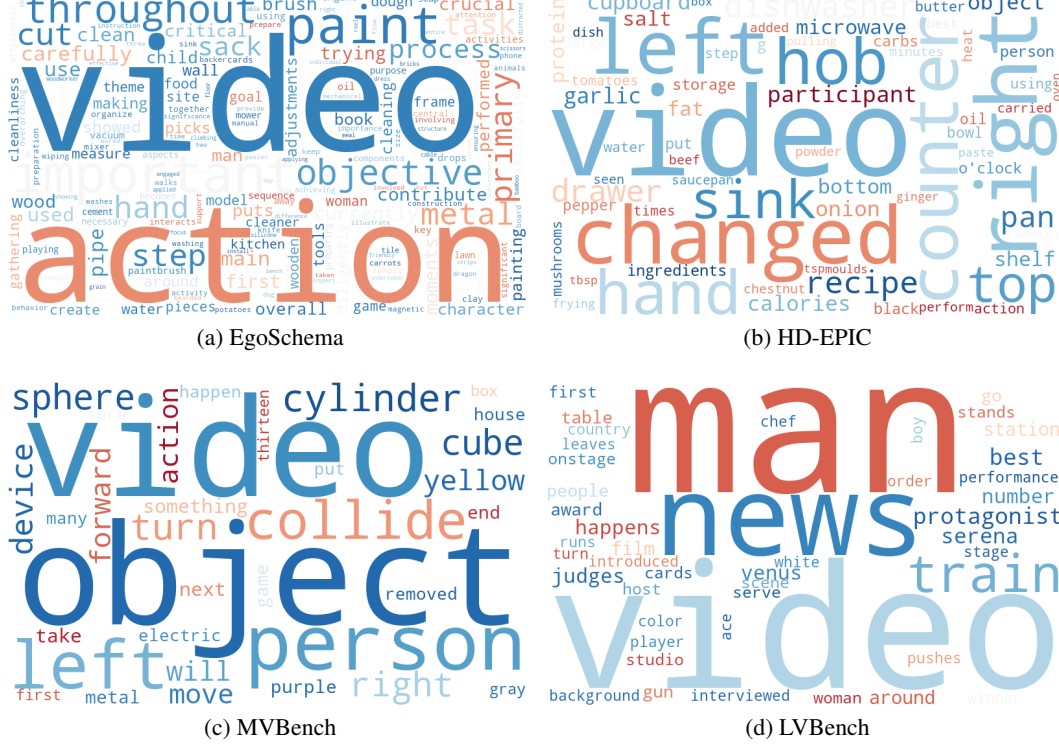

Figure 18: InternVideo word clouds showing the frequency of the words within the dataset subset via their size and the attribution by their colour, **blue** for positive attribution and **red** for negative.

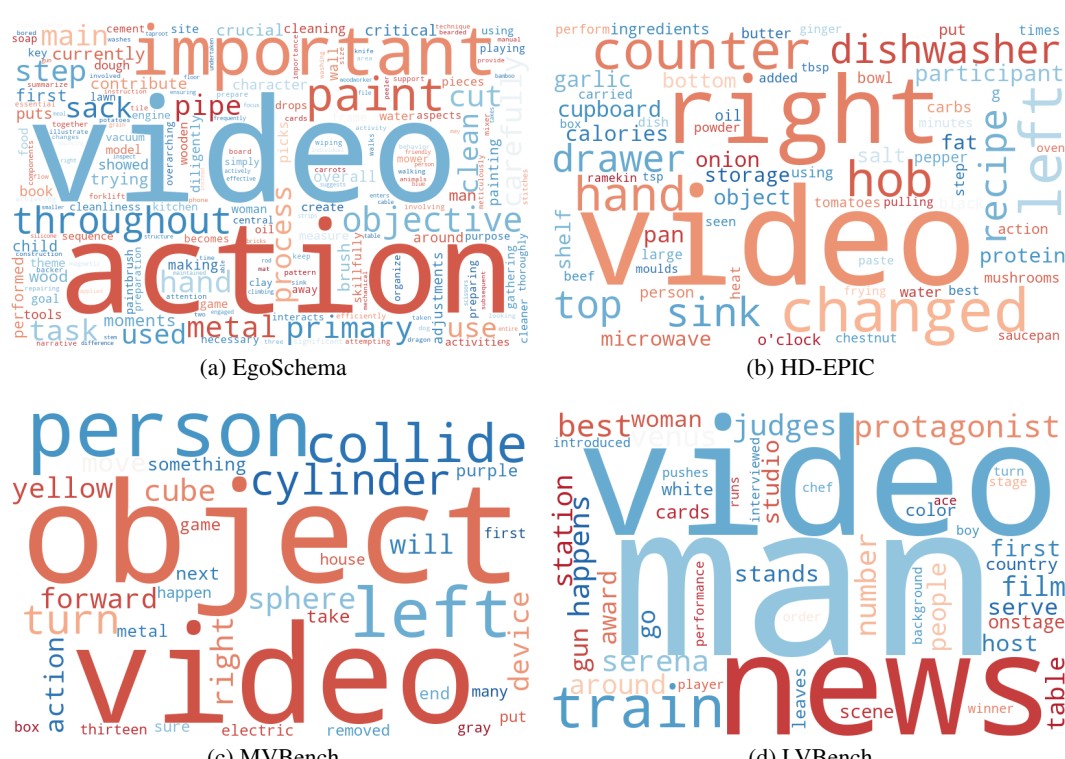

(a) EgoSchema

(b) HD-EPIC

(c) MVBench

(d) LVBench

Figure 19: VideoLLaMA2 word clouds showing the frequency of the words within the dataset subset via their size and the attribution by their colour, **blue** for positive attribution and **red** for negative.

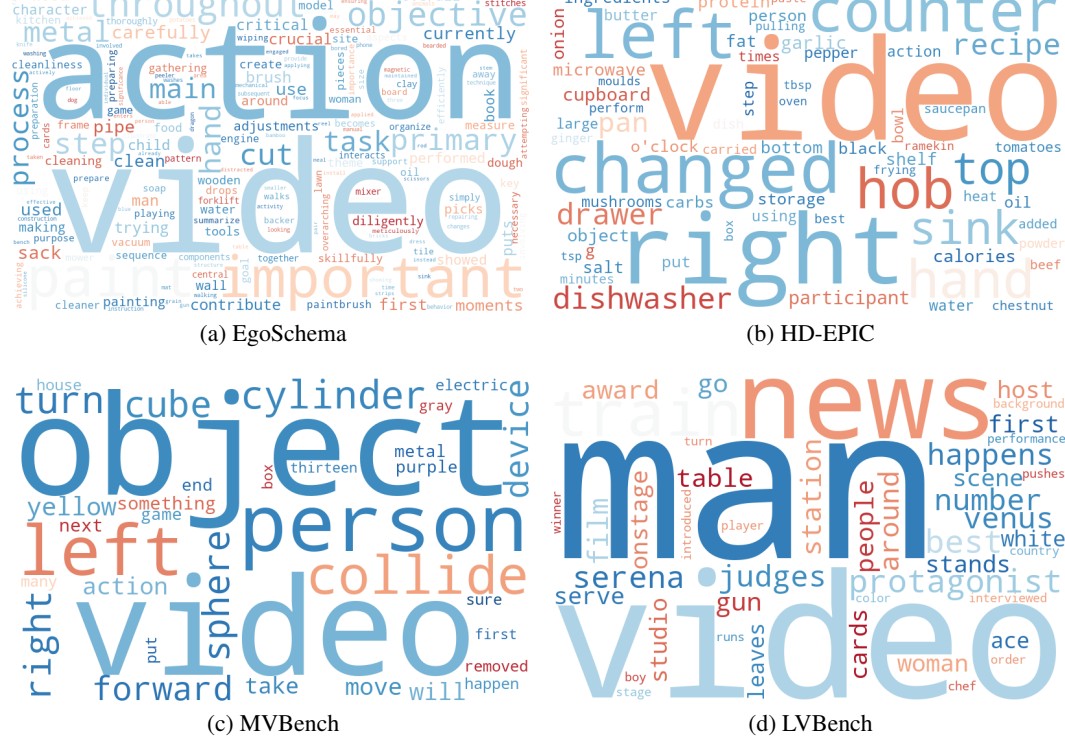

(a) EgoSchema

(b) HD-EPIC

(c) MVBench

(d) LVBench

Figure 20: LLaVA-Video word clouds showing the frequency of the words within the dataset subset via their size and the attribution by their colour, **blue** for positive attribution and **red** for negative.

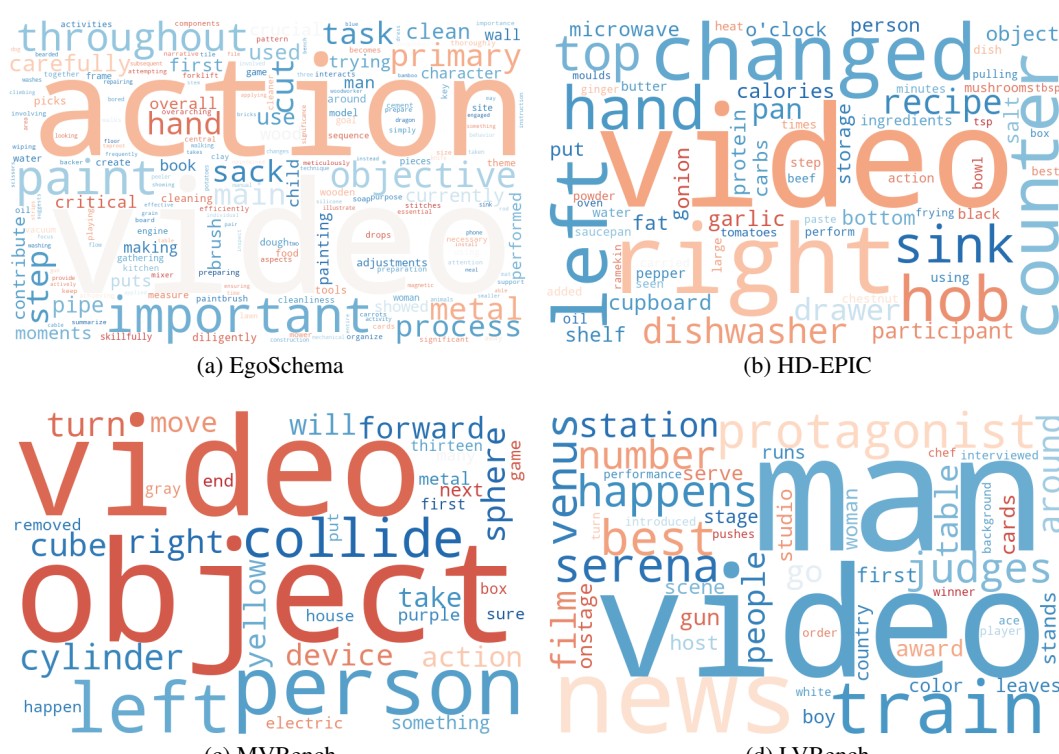

(a) EgoSchema

(b) HD-EPIC

(c) MVBench

(d) LVBench

Figure 21: LongVA word clouds showing the frequency of the words within the dataset subset via their size and the attribution by their colour, **blue** for positive attribution and **red** for negative.

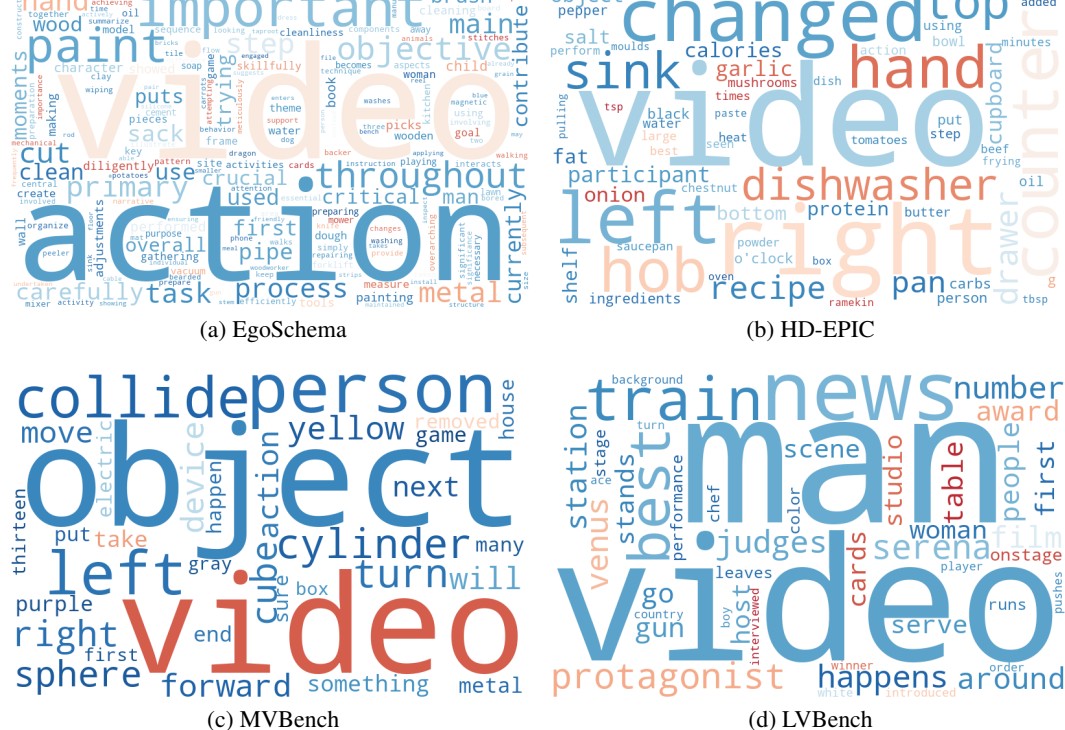

(a) EgoSchema

(b) HD-EPIC

(c) MVBench

(d) LVBench

Figure 22: VideoLLaMA3 word clouds showing the frequency of the words within the dataset subset via their size and the attribution by their colour, **blue** for positive attribution and **red** for negative.

