# OpenReview forum: "A Video Is Not Worth a Thousand Words"
_ICLR.cc/2026/Conference — Submitted to ICLR 2026_

### Official Review · Reviewer_HatP · 2025-10-26

**Soundness:** 2
**Presentation:** 2
**Contribution:** 2
**Rating:** 4
**Confidence:** 2

**Summary:**

This paper examines whether video–language models (VLMs) genuinely utilize visual information when performing multiple-choice video question answering (VQA), or whether their predictions are predominantly text-driven. To this end, the authors propose a Shapley-value–based attribution framework that operates on arbitrarily grouped features, such as entire video frames and textual components, to quantify both per-feature and modality-level contributions. These contributions are normalized to ensure independence from model accuracy and modality length, yielding two interpretable metrics: Modality Contribution and Per-Feature Contribution. Using this framework, the authors analyze six open-source VLMs (FrozenBiLM, InternVideo, VideoLLaMA2, LLaVA-Video, LongVA, and VideoLLaMA3) across four representative benchmarks (EgoSchema, HD-EPIC, MVBench, and LVBench). The findings reveal that: (1) VLMs systematically underutilize video information relative to text, (2) many questions can be answered reasonably well without the question text, and (3) increasing the number of answer options tends to reduce text dominance and encourage greater reliance on visual and question modalities, suggesting that current multiple-choice VQA often emphasizes distractor elimination rather than genuine multimodal reasoning.

**Strengths:**

1. The introduction of a Shapley-value–based attribution framework to investigate modality bias in VLMs is novel. This approach provides a principled and interpretable means of quantifying the relative contributions of visual and textual inputs, which has not been systematically explored in prior multimodal reasoning research.

2. The paper presents an extensive and well-structured empirical study, evaluating six VLMs with varying context lengths across four diverse VQA datasets that differ in perspective (egocentric vs. exocentric) and video duration (short vs. long).

3. The empirical findings, namely, that (1) video information is consistently underutilized compared to textual cues, (2) many questions can be answered even without the question text, and (3) increasing the number of answer options mitigates textual dominance, offer valuable insights into the limitations of current multiple-choice VQA formulations and provide meaningful guidance for future multimodal reasoning research.

**Weaknesses:**

1. Despite the thorough experimental analysis, the paper lacks a clear methodological or algorithmic contribution that meets the technical novelty threshold typically expected at ICLR. The proposed attribution framework, while well-motivated, primarily extends existing interpretability techniques rather than introducing a fundamentally new learning paradigm or model architecture to address their findings.

2. Many of the reported empirical findings reiterate observations that have been discussed in prior literature and thus offer limited novelty. For instance, the underutilization of visual information in VQA systems has been previously discussed in [1, 2] and the observation that increasing the number of answer options leads to decreased performance or altered modality reliance is largely intuitive.

3. As acknowledged by the authors, the study’s scope is restricted to multiple-choice VQA, which limits the generality of its conclusions. It remains unclear whether the same modality attribution patterns would hold for open-ended VQA, captioning, or other multimodal reasoning tasks, thereby constraining the broader applicability of the findings.

[1] Large Language Models are Temporal and Causal Reasoners for Video Question Answering

[2] Generative Bias for Robust Visual Question Answering

**Questions:**

N/A

---

> ### Author Response · Authors · 2025-11-20
> **Response**
>
> # Weakness 1
>
> We don’t endeavour to solve this problem, rather, we provide a principled method that may be applied to any theoretical combination of model and dataset to study these emerging problems and provide a comprehensive analysis. Currently, it is very difficult to quantify whether works are improving the fusion and use of modalities because the only general metric for comparison discussed is model accuracy. Not only have we exposed the extent to which long video is under-exploited, we’ve also described a framework that allows for case study and fair comparison between future works.
>
> # Weakness 2
>
> The topic has been explored but almost always in the context of metrics derived from accuracy, which is not a rigorous measure of the input contribution towards model output. In fact, [1] concludes that models over-rely on the question modality, but our results directly contradict this. We’ve demonstrated that answer contributions are significantly larger, and that models are often completely unaffected when questions are masked (see Table 2 in main paper). The datasets examined in [1] are significantly less complex than the datasets we studied (see Appendix Section D), and include very short video context which is ~1.5 minutes long at the extreme compared to ~7.2 hours in HD-EPIC. Furthermore, the analysis in [2] is limited to image question answering which is a very different domain to video VQA because it lacks the temporal dimension. We believe that our results remain valuable as they’re derived from a mathematically rigorous method of feature attribution that is under-used when approaching interpretability of these new and more complex models/datasets.
>
> # Weakness 3
>
> Multiple-choice VQA is a ubiquitous benchmark style with a more rigorously defined evaluation setup than open-ended VQA (captioning metrics vs. LLMs as judge). The LLM as judge approach is essentially uninterpretable and it’s been demonstrated that their reliability is task dependent [1]. However, it is trivial to extend our setup to an open-ended scenario by utilising any arbitrary captioning metric. To test this, we have calculated Shapley values using ROUGE-L as the reward between the predicted text and the ground truth text (for an open-ended version of EgoSchema). These results have been added to the updated PDF in Appendix Section E.1:
>
> |  | Modality Contribution - V | Modality Contribution - Q | Per-Feature Contribution - V | Per-Feature Contribution - Q |
> |---|---|---|---|---|
> | LLaVA-Video | 0.38 | 0.62 | 0.21 | 0.79 |
> | VideoLLaMA3 | 0.69 | 0.31 | 0.25 | 0.75 |
>
> We see that the ratio of contribution between video and question is the same (if not more skewed towards the text) as in the original multiple-choice results. This, along with the low Per-Feature Contributions, demonstrates that even without the leading information present in answer choices, the video modality lacks contribution. Captioning is not a similarly relevant task as it only requires one input modality (video).
>
> [1] Bavaresco, Anna et al. “LLMs instead of Human Judges? A Large Scale Empirical Study across 20 NLP Evaluation Tasks.” Annual Meeting of the Association for Computational Linguistics (2024).

---

### Official Review · Reviewer_8H6M · 2025-10-30

**Soundness:** 4
**Presentation:** 4
**Contribution:** 2
**Rating:** 4
**Confidence:** 4

**Summary:**

The MM-SHAP paper demonstrated the use of Shapley values in order to measure the relative importance of image and text modalities for VQA, and showed that, between images and text, mode collapse might happen in either direction.  This paper extends the same methods to long videos.

**Strengths:**

The extension to video generates new results that differ from the image+text results in somewhat interesting ways.  Specifically: (1) unlike images, video is always less important than text, though only a small number of questions can be correctly answered with the video completely masked, (2) the importance of the video increases if the number of candidate answers in a multiple-choice question is increased by rotating in some answers randomly chosen from other questions.

**Weaknesses:**

Although the analyses of video are interesting, it is difficult to recommend acceptance because all of the proposed algorithms have previously been applied, in more or less the same form, to image-text VQA.

**Questions:**

The last paragraph of section 3.2 defines the reward to be "the logits of the predicted text tokens."  I first read that to say that you use the logits of the answer generated by the VLM, but if that were true, it would be impossible to measure \phi^{gt} unless the VLM generated the correct answer, right?  I think you must be measuring the logit of the GT answer, even if the VLM does not choose to generate the GT answer?

---

> ### Author Response · Authors · 2025-11-20
> **Response**
>
> # Weakness 1
>
> Image patches are very different to video frames, as is the domain of image VQA and video VQA. With video there is the added dimension of time, which quite drastically increases the possible complexity of the question space. As well as this, we’ve benchmarked models and datasets that do not overlap with previous works significantly. Our intent was to rigorously quantify modality contribution (or lack of it), and our results imply that there is a greater textual bias in the studied domain and have highlighted the necessity of metrics that retain meaning as modality sizes differ.
>
> # Question 1
>
> Each predicted text token can be returned with the corresponding logits for *all* classes in the tokenizer vocabulary. You are correct, we extract the logits from all of the answer choices (A, B, …, E) simultaneously from the predicted text token. This approach is similar to that of [1], where Shapley values are calculated for all classes jointly. This way, we can produce attributions for the ground truth class even when the model predicts a false answer.
>
> [1] Price, Will, and Dima Damen. “Play Fair: Frame Attributions in Video Models.” ACCV (5), vol. 12626, Springer, 2020, pp. 480–97.

---

### Official Review · Reviewer_Wd6d · 2025-10-31

**Soundness:** 3
**Presentation:** 3
**Contribution:** 3
**Rating:** 6
**Confidence:** 3

**Summary:**

Brief Summary: This is an investigative / diagnostic paper to find how much does the video modality in general contribute for VLM understanding, and authors primarily focus on the VQA task. This is somewhat known intuitively that video plays a downsized role, but the authors provide a framework based on multi-modal shapely values to find per-modality and per-feature contributions for an answer. Experiments are conducted over 4 benchmarks (Egoschema, hd-epic, mvbench, lvbench) which shows other than video-llama3, video contribution is quite low.

**Strengths:**

Pros:
1. The paper is very well motivated from an evaluation perspective. Finding which modality plays part in final output is very important. I can see many downstream tasks requiring explainability.

2. In a way, the authors are under-selling the paper as a diagnosis for methods only. I think especially from L299, this can be a good diagnostic for benchmark creation as well. Usual way for benchmark creation to ensure requirement of video modality is to show video-blind models perform similar to random. The proposed method is essentially taking it a step further by doing per-modality and per-feature representations.

3. The authors provide reasonably detailed experiments including in the supplementary. Authors experiment on multiple datasets with diverse domains (ego and exo videos) and models frozen-bilm, intern-video, video-llama2/3, llava-video, longva.

4. The gemini-oracle experiments are quite interesting, in that the frames disagree with each other.

**Weaknesses:**

Cons:

1. My main concern is that the only task evaluated in multiple-choice VQA. This severely restricts it applications (which authors also note in limitations section 5). Authors should at least experiment with full-string match? I am slightly confused why a trivial extension of proposed method cannot be done with say removing parts of the output tokens? It would be great to have the authors expand on this.

2. The main takeaway is that video modality is under-represented. It is a good to have empirical evidence but is not super surprising. Also, quite confused, why image-language tasks are not considered here? Some direct comparison with MM-SHAP would be interesting to see.

3. In usual VLM, an important aspect is that of tool calling. How is tool-calling being handled via this method?

4. (Minor) I think it also makes sense to investigate even more modalities outside of video, such as audio or IMU as done in image-bind kind of works. It would significantly strengthen the paper.

5. In the shapely value computation, the authors choose to zero-out the video and text is replaced with whitespace, but this design choice is not really justified. An alternative would be to simply remove the frame in the video and remove the token in text? (unless i missed some ablation experiment).


----

Overall: Rating 6/10

The overall idea mostly makes sense that we can use shapely values to estimate video influence in downstream task. The paper has numerous mostly well-thought out experiments which ought to be valuable for the VLM community in general. Main concern is that the approach is only used for MCQ setting which is too narrow. The paper could be improved by adding more details/discussion on how long-context, tool calling, black-box setting, additional baselines, additional modality comparisons.

**Questions:**

Q1. How does the behavior change in long-context settings, such as going to hour-long videos which nearly fills the context window.

Q2. Can the method be extended to black-box models such as gpt? Right now, we are using logit output, but is there a way to avoid using pure logit information, say via monte-carlo (via pass@K) ? Not sure.

Q3. The evaluation by itself is quite expensive (as noted in appendix C). Is the 7200 gpu hours for each experiment or across all experiments? Is there some parallelization that can be done? Otherwise, it is not really of practical use at the moment.

---

> ### Author Response · Authors · 2025-11-20
> **Response Part 1**
>
> # Strength 2
>
> Thank you for pointing this out. We feel like this is one of the main strengths of the method, as the trend in recent years has been to invent specific datasets to test specific facets of models, or to use strong models to verify that a dataset is “difficult”. However, normalised modality contributions are easily cross-comparable and can be an early indicator that a newly gathered dataset contains less shortcuts. We will update the introduction and conclusion to reflect this.
>
> # Weakness 1
>
> In this paper we intentionally focus on multiple-choice VQA because of the greater inherent textual bias present in the inputs. The evaluation of open-ended VQA evaluation is still somewhat of an open question, traditionally being solved via captioning metrics and more recently with LLMs as judges. The LLM as judge approach is essentially uninterpretable and it’s been demonstrated that their reliability is task dependent [1]. However, it is possible to extend our method of feature attribution to open VQA by replacing the scoring function with a captioning metric between the predicted and ground truth text. Any captioning metric could be employed, or even similarity in a text encoder space. Applying this to EgoSchema (with the dataset trivially altered to become open question-answer) and using ROUGE-L as the Shapley reward, we get the following results:
>
> |  | Modality Contribution - V | Modality Contribution - Q | Per-Feature Contribution - V | Per-Feature Contribution - Q |
> |---|---|---|---|---|
> | LLaVA-Video | 0.38 | 0.62 | 0.21 | 0.79 |
> | VideoLLaMA3 | 0.69 | 0.31 | 0.25 | 0.75 |
>
> We can see that the Modality Contributions appear large for video, but it’s worth noting that the modalities have significantly different sizes (180 frames vs. ~30 words), which is demonstrated by the lacking Per-Feature Contributions. Overall, the ratio between video and question is similar to that for multiple-choice VQA. These results have been added to the updated PDF in Appendix Section E.1, but we leave a more in depth exploration of the topic as future work.
>
> # Weakness 2
>
> MM-SHAP quantifies feature contribution towards a predicted text caption, as opposed to the classification scenario we studied with multiple-choice where we can use ground truth/negative logits. Furthermore, they mask features at the token level, whereas we do masking at an arbitrary scale (focusing on frames and words). We don’t focus on image/language tasks because we are interested in the importance of the temporal dimension, and how video encoders often struggle to capture temporal and causal relationships. There are image VQA examples present for certain question types in HD-EPIC when the task depends on single frames.
>
> # Weakness 3
>
> The open source VLMs we benchmarked do not tend to make use of tool calling even if the feature is present in the LLM. We just consider any potential tool call part of the black box model. If, however, you want to quantify whether inputs contribute towards a tool call, you could theoretically use the models’ metrics for picking tools as reward for the Shapley value calculation.
>
> # Weakness 4
>
> The extension to other modalities is trivial but we wanted to focus on textual bias because of the obvious Clever Hans effect and the danger it poses when language models are used in vision models. Additionally, these open source video language models are not intended to be used with audio and in fact would require modification and training for them to do so. Our approach can of course be applied to any theoretical ensemble of modalities, but we are particularly interested in evaluating progress in the video domain.
>
> # Weakness 5
>
> Shapley coalitions require querying the model with entirely masked input and not all models support the notion of true “empty” input (FrozenBiLM and InternVideo take the product of video and text features for example). Therefore, we decided to mask in this manner as it is applicable universally across all of the models and does not introduce differences in the expected input distribution that may affect the output.
>
> [1] Bavaresco, Anna et al. “LLMs instead of Human Judges? A Large Scale Empirical Study across 20 NLP Evaluation Tasks.” Annual Meeting of the Association for Computational Linguistics (2024).

---

> > ### Author Response · Authors · 2025-11-20
> > **Response Part 2**
> >
> > # Question 1
> >
> > If there are enough frames to sample then we always fill up the context window for all models. HD-EPIC contains a large range of video context lengths (from single frames to seconds to several hours), so we’ve plotted a graph of the video Per-Feature Contribution against video length for this subset. The plot can now be found in Appendix Section E.6 of the updated PDF. We see that shorter video input tends to have larger contributions and longer video input tends to have much smaller contributions, implying that long context video is *not* being integrated well by the model. In fact, the Pearson correlation between the two variables is -0.36, which shows a slight negative correlation.
> >
> > # Question 2
> >
> > If the API supports logit output, it can be applied as is. If the API *doesn’t* support logit output, as we’ve demonstrated in Weakness 1, we can use the ROUGE-L method to avoid using logits altogether for the open VQA scenario as it just requires a predicted output. This can similarly be applied to the multiple-choice scenario for black-box models, where the ROUGE scoring function will turn into a one-hot vector for the answer choices (A, B, …, E). We will add a discussion of this to future work.
> >
> > # Question 3
> >
> > This value is across *all* experiments in the paper and is significantly inflated by the answer replacement experiments as they required us to calculate Shapley values for 5 more versions of each dataset with altered annotations. The computation required to calculate Shapley values for a given model/dataset combination is considerably smaller. For example, VideoLLaMA3 on HD-EPIC used ~70 GPU hours.

---

> ### Comment · Reviewer_Wd6d · 2025-11-26
>
> I thank the authors for their rebuttal. Most of my comments are as addressed, hence I would like to keep my score of 6/10.
>
> Additional experiments with other tasks, including tool calling would be interesting to consider and would significantly strengthen the paper.

---

### Meta-Review · Area_Chair_UpU1 · 2026-01-08

**Summary:**

This paper proposes a Shapley-value-based attribution framework to quantify the extent to which VLMs, use visual information for video QA tasks.

The reviewers appreciate the novel framework for investigating modality bias.

The concerns raised by the reviewers include
- lack of methodological or algorithmic contribution
- much of the reported empirical findings are already well-established in the literature
- limited scope of study (multiple choice QA only, video only instead of image as well)

**Reviewer Concerns:**

The rebuttal mostly justifies existing choices in the work, rather than address the overall weaknesses.  While some small experiments e.g. on open-ended QA are added, it avoids addressing the main concern, which is the fact that much of the reported findings are already well-established in the literature.  The claim that video is fundamentally different than images is not well-supported either.

**Reviewer Scores:**

Wd6d  - score of 6, reviewer said they want to keep their score in the discussion
8H6M - score of 4, I don't think this reviewer would further raise the score
HatP - score of 4.  I don't think this reviewer would further raise the score

---

### Decision · Program_Chairs · 2026-01-26

Reject